# RGLA: REVERSE GRADIENT LEAKAGE ATTACK US-ING INVERTED CROSS-ENTROPY LOSS FUNCTION

## ABSTRACT

Federated learning (FL) has gained widespread adoption due to its ability to jointly train models by only uploading gradients while retaining data locally. Recent research has revealed that gradients can expose the private training data of the client. However, these recent attacks were either powerless against the gradient computed on high-resolution data of large batch size or often relied on the strict assumption that the adversary could control and ensure unique labels for each sample in the attacked batch. These unrealistic settings and assumptions create the illusion that data privacy is still protected in real-world FL training mechanisms. In this paper, we propose a novel gradient leakage attack named RGLA, which effectively recovers high-resolution data of large batch size from gradients while considering duplicate labels, making it applicable in realistic FL scenarios. The key to RGLA is to invert the cross-entropy loss function to obtain the model output corresponding to the private model inputs. Next, RGLA directly computes the feature map inputted into the last fully-connected layer leveraging the obtained model output. Finally, a previous generative feature inversion model is used to invert the feature map of each sample to model input space. Extensive experimental results demonstrate that RGLA can reconstruct $224 \times 224$ pixels images with a batch size of 256 while considering duplicate labels. Our source code is available at `https://github.com/AnonymousGitHub001/RGLA`.

## 1 INTRODUCTION

Federated Learning (FL) (McMahan et al., 2017; Shokri & Shmatikov, 2015) safeguards the privacy of client data by only uploading gradients without sharing the actual data. However, the emergence of Gradient Leakage Attacks (GLAs) shredded the good impression of FL by stealing clients' private data. Broadly, GLAs can be classified into two main types: optimization-based attacks (Zhu et al., 2019; Geiping et al., 2020) and analytics-based attacks (Phong et al., 2017; Zhu & Blaschko, 2020; Fowl et al., 2021; Xue et al., 2023). The former optimizes the dummy data and dummy label to minimize the distance between its gradient and the observed gradient. When this distance is small enough, the dummy data can be considered as reconstructed data. The latter directly derives the reconstructed data from the gradient through a series of analytical equations.

However, these two types of GLAs may struggle to work in realistic FL settings. Optimization-based attacks are limited to unrealistic small-batch or low-resolution data. In fact, high-resolution data of large batch size is more common in FL. Nevertheless, as heightened by Zhu et al. (2019), high resolution or large batch size makes the optimization more difficult for there are more variables to solve during optimization. Additionally, we notice that analytics-based attacks frequently rely on unrealistic assumptions, such as the gradient computed on a single sample (Zhao et al., 2020a; Phong et al., 2017) and each label in a batch is unique (Zhu & Blaschko, 2020; Xue et al., 2023). Unfortunately, as highlighted by recent studies, executing GLAs on data containing duplicate labels presents a significant challenge (Zhu & Blaschko, 2020; Fowl et al., 2021). For instance, Fowl et al. (2021) emphasizes that analytics-based attacks fail to attack once the duplicate labels occur. Fig. 1 illustrates the ineffectiveness of optimization-based methods, represented by Geiping et al. (2020), and analytics-based methods, represented by Xue et al. (2023), in such a simple FL setting.

To address the above challenges, we introduce *Reverse Gradient Leakage Attack* (RGLA), a novel GLA that effectively reconstructs high-resolution data of large batch size without assuming any

| Original batch of 224×224 pixels, ground-truth | RGLA(Ours), **PSNR↑=19.91**, time:14.89(s) |

| IG(NeurIPS'20), PSNR↑=12.06, time:4993.16(s) | FGLA(INFOCOM'23), PSNR↑=13.91, time:1.21(s) |

Figure 1: Visualization of RGLA and other state-of-the-art GLAs on high-resolution data in a setting containing duplicate labels. The data in the red boxes have the same labels.

specific label distribution. The RGLA is inspired by the observation that gradients can leak the model output corresponding to the model input, and subsequently, the leaked model output reveals the data itself (Zhu & Blaschko, 2020). RGLA consists of three steps. Specifically, we first propose three innovative optimization objectives, inverting the cross-entropy loss function of the target model to reconstruct the corresponding model output logits for each sample through optimization. Next, the reconstructed logits are used in an analytical attack to directly compute the feature map inputted into the last fully-connected layer (FCL) of the target model for each data input. Finally, prior work (Xue et al., 2023) is employed to generate the data based on their feature maps with no effort.

Compared to optimization-based methods, RGLA requires less than $\frac{1}{200}$ of their time, demonstrates robustness to high resolution, and can tackle any batch size providing that the number of classes is higher than the batch size. Compared to analytics-based methods, RGLA is the first to be able to handle label duplication. Our contributions can be summarized as follows:

- We propose RGLA for performing a gradient leakage attack in FL. RGLA establishes equations for every value in the original model output and optimizes the model output, thus addressing the challenge of reconstructing private data with duplicate labels. As a result, RGLA pushes the boundary of GLAs towards more practical FL scenarios.

- RGLA optimizes the model output, with its output being positively correlated only with the batch size and the number of classes, independent of pixel size. This enables achieving a more efficient optimization within a smaller optimization space. Consequently, it addresses the challenge of reconstructing high-resolution data.

- Extensive experiments show that RGLA outperforms state-of-the-art methods and exhibits robustness to various defenses, label distributions, and batch sizes. RGLA can rapidly reconstruct data batch of $224 \times 224$ pixels images in a batch size of 256 within 30 seconds, thereby proving that gradients can still leak information in more complex FL settings.

## 2 RELATED WORKS

Existing GLAs for reconstructing private data can be broadly classified into two categories: optimization-based attacks and analytics-based attacks. Table 1 briefly compares RGLA with existing methods and showcases RGLA's superior performance across these assessed parameters.

**Optimization-based methods** optimize dummy data for the generated gradient to be as close as possible to the observed gradient. In more recent studies, regular terms may be added to the dummy data. Zhu et al. (2019) introduced DLG (Deep Leakage from Gradients), the first reconstruction of the private data, but only works on low-resolution images of 32×32 pixels and batch sizes of 8. Zhao et al. (2020a) improved DLG by revealing the ground-truth label through FCL gradient signs but only works on a single data. Geiping et al. (2020) (Inverting Gradients, IG) used cosine similarity as a distance metric and total variation regularization for image reconstruction. Yin et al. (2021) further improved DLG using multiple optimization terms, including the standard image priors (Mahendran & Vedaldi, 2015; Mordvintsev et al., 2015) that penalize the total variance and $l_2$ norm of the dummy image as well as using strong priors BatchNorm statistics. Jeon et al. (2021) optimized low-dimensional features and generator to fully utilize a pre-trained generative model to invert gradient. Based on Jeon et al. (2021), Li et al. (2022) only optimized low-dimensional features to invert the gradient. Yang et al. (2022) (Highly Compressed Gradients Leakage Attack, HCGLA) investigated new dummy data initialization for better quality. Yue et al. (2022) explored an image reconstruction attack at a semantic level and measured the corresponding image privacy leakage. Wu et al. (2023) trained a model to learn the mapping between the gradient and corresponding inputs.

Table 1: A comparison of RGLA with existing methods based on criteria such as batch size, image size, the number of classes $C$, speed, defense mechanisms, and data label with duplicate label.

| | Max Batch Size $\times$ Image Size | Class Count $> 2$ | Fast Speed | Strict Defense[1] | Data with Duplicate Label |
|---|---|---|---|---|---|
| Zhu et al. (2019) | $8 \times (32 \times 32)$ | ✓ | ✗ | ✗ | ✓ |
| Geiping et al. (2020) | $100 \times (32 \times 32)$ | ✓ | ✗ | ✗ | ✓ |
| Jeon et al. (2021) | $48 \times (64 \times 64)$ | ✓ | ✗ | ✗ | ✓ |
| Yang et al. (2022) | $1 \times (32 \times 32)$ | ✓ | ✗ | ✗ | ✓ |
| Li et al. (2022) | $1 \times (224 \times 224)$ | ✓ | ✗ | ✗ | ✓ |
| Phong et al. (2017) | $1 \times (32 \times 32)$ | ✓ | ✓ | ✗ | ✗ |
| Fan et al. (2020) | $8 \times (32 \times 32)$ | ✓ | ✓ | ✗ | ✗ |
| Zhu & Blaschko (2020) | $1 \times (32 \times 32)$ | ✗ | ✓ | ✗ | ✗ |
| Fowl et al. (2021) | $64 \times (224 \times 224)$ | ✓ | ✓ | ✗ | ✗ |
| Xue et al. (2023) | $256 \times (224 \times 224)$ | ✓ | ✓ | ✗ | ✗ |
| RGLA(Ours) | $(C{+}1) \times (224 \times 224)$ | ✓ | ✓ | ✓ | ✓ |

[1] Strict defense refers to a highly compressed gradient with a compression ratio of 99.9%, noisy gradient with $\sigma = 0.01$, and clipped gradient with $\mathcal{C} = 4$.

**Analytics-based methods** compute private data directly using analytical formulas associated with the gradient. Phong et al. (2017) introduced a formula $\boldsymbol{f}' = \nabla\boldsymbol{\theta}_L^W / \nabla\boldsymbol{\theta}_L^b$ to recover training data, where $\nabla\boldsymbol{\theta}_L^W$ denotes the gradient of the weight term in $L^{th}$ layer (also FCL) in the model and $\nabla\boldsymbol{\theta}_L^b$ denotes the gradient of the bias term in $L^{th}$ layer in the model. Fan et al. (2020) expanded the method of Phong et al. (2017). to handle multiple-layer perceptions. R-GAP (Zhu & Blaschko, 2020) provided a closed-form recursive procedure to recover data from gradients. Fowl et al. (2021) retrieved portions of data of batch size 64 by modifying the model structure, but it can be easily detected. Boenisch et al. (2021) proposed the "trap weights" method to augment the number of neurons in the model, enhancing the possibility of directly computing training data from the gradient. Xue et al. (2023) (Fast Gradient Leakage Attacks, FGLA) used an analytical method to separate the feature map of original data from gradients and fed it into a pre-trained generator, enabling rapid reconstruction of a data batch. More recent work (Cocktail Party Attack (Kariyappa et al., 2023), CPA) treats the gradient inversion as a Blind Source Separation (BSS) problem and then CPA uses independent component analysis (ICA) to address the BSS problem.

## 3 THREAT MODEL

Consider an FL system consisting of two entities, a parameter server $\mathcal{A}$, and multiple clients. An honest but curious parameter server $\mathcal{A}$ is assumed to be an adversary who aims to steal the client's data while following the FL protocol. The target model in our attack is the convolutional neural network with the fully-connected layer as the last layer and trained by the cross-entropy loss function, as such setting is widely adopted for image classification and widely assumed as the target model in most existing methods (Zhu & Blaschko, 2020; Fowl et al., 2021; Xue et al., 2023). Similar to most existing works (Yin et al., 2021; Xue et al., 2023), our attack primarily occurs during the initialization or pre-training phase of model training and focuses on the gradient of one iteration. As a server, $\mathcal{A}$ has access to the parameters of the global model, including weights ($\boldsymbol{\theta}^W$) and biases ($\boldsymbol{\theta}^b$), as well as the gradients uploaded by each client during training. $\mathcal{A}$ is assumed to have access to auxiliary datasets as it is widely used by the prior works (Liu et al., 2021; Zhao et al., 2020b; Li et al., 2022; Xue et al., 2023). Previous research (Wainakh et al., 2021; Ma et al., 2022) has demonstrated that the label information can be independently recovered with high accuracy (up to 100%), so it is assumed that $\mathcal{A}$ can obtain the ground-truth labels. Similar to Xue et al. (2023), the target model is assumed to omit global pooling layers for more information from the gradient. Finally, considering a realistic FL scenario, $\mathcal{A}$ is assumed not to get access to batch norm statistics of the client's data.

## 4 REVERSE GRADIENT LEAKAGE ATTACK

### 4.1 MOTIVATATION: REVERSE DISAGGREGATION AND FAST GENERATION

To better understand our approach, let's look at two simple examples that inspired us. Let us begin by understanding the common processing data flow of a convolutional neural network. A data

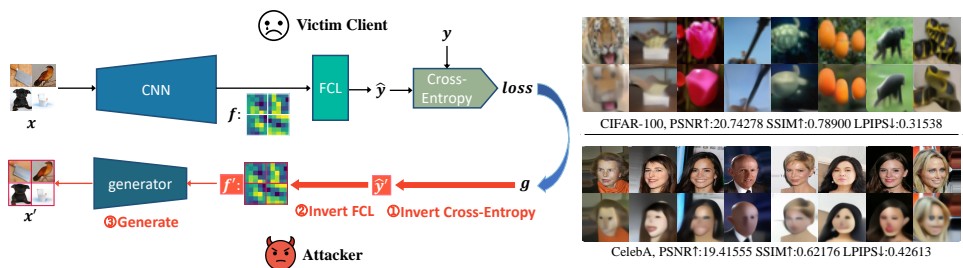

Figure 2: **Left**: The overview of our proposed RGLA. **Right**: The statistical metric and visualization of reconstruction results on the CIFAR-100 and CelebA datasets.

---

**Algorithm 1** Private data leakage from the captured gradients.

**Input:** $\boldsymbol{\theta}$: the global model parameters; $\nabla\boldsymbol{\theta}$: gradients calculated by training data; $\mathcal{D}_{aux}$: the auxiliary dataset to train the generator; Ground-truth label: $\boldsymbol{y}$.
**Output:** private training data $\boldsymbol{x}'$

1: **procedure** RGLA($\boldsymbol{\theta}$, $\nabla\boldsymbol{\theta}$, $\boldsymbol{y}$, $\mathcal{D}_{aux}$)
2:     $\hat{\boldsymbol{y}}_1' \leftarrow \mathcal{N}(0,1)$                    ▷ Initialize dummy model outputs $\hat{\boldsymbol{y}}'$.
3:     **for** $i \leftarrow 1$ to $Epochs$ **do**
4:         $\mathcal{L}oss_{total}^i \leftarrow \alpha_1\mathcal{L}_w(\hat{\boldsymbol{y}}_i') + \alpha_2\mathcal{L}_b(\hat{\boldsymbol{y}}_i') + \alpha_3\mathcal{L}_{loss}(\hat{\boldsymbol{y}}_i')$
5:         $\boldsymbol{y}_{i+1}' \leftarrow \boldsymbol{y}_i' - \eta\nabla_{\boldsymbol{y}_i'}\mathcal{L}oss_{total}^i$       ▷ Update the dummy model outputs $\hat{\boldsymbol{y}}'$.
6:     **end for**
7:     $\boldsymbol{f}' = (\partial l\,(\hat{\boldsymbol{y}}', \boldsymbol{y})/\partial\hat{\boldsymbol{y}}')^{-1} \cdot \nabla\boldsymbol{\theta}_L^W$              ▷ Compute the dummy feature map $\boldsymbol{f}'$.
8:     $\mathcal{G} \leftarrow$ Train a generator by inputting feature maps and outputs corresponding images using model parameters $\boldsymbol{\theta}$ and auxiliary dataset $\mathcal{D}_{aux}$.
9:     $\boldsymbol{x}' \leftarrow \mathcal{G}(\boldsymbol{f}')$                    ▷ Generate the reconstructed data.
10:     **return** $\boldsymbol{x}'$
11: **end procedure**

---

batch $\boldsymbol{x} \in \mathbb{R}^{\text{B}\times\text{Channel}\times\text{Hight}\times\text{Width}}$ input into the convolutional neural network is first processed by a convolutional layer of the network, producing a feature map $\boldsymbol{f} \in \mathbb{R}^{B\times H}$. $\boldsymbol{f}$ is then processed by a FCL using the equation $\hat{\boldsymbol{y}} = \boldsymbol{f} \cdot \boldsymbol{\theta}_L^W + \boldsymbol{\theta}_L^b$ thus yielding a model output logits $\hat{\boldsymbol{y}} \in \mathbb{R}^{B\times C}$. Finally, the model output $\hat{\boldsymbol{y}}$ and the ground-truth labels $\boldsymbol{y} \in \mathbb{R}^{B\times 1}$ are input into the cross-entropy loss function to get the $loss$. Where $\cdot$, $B$, $H$, and $C$ denote the matrix multiplication, batch size, the number of neurons in the input layer, and the number of classes, respectively.

As demonstrated by Zhu & Blaschko (2020); Fowl et al. (2021); Wainakh et al. (2021); Xue et al. (2023), the feature map $\boldsymbol{f}$ input into the last FCL can be revealed approximately by:

$$\boldsymbol{f}_{\boldsymbol{y}(i)} \approx \nabla\boldsymbol{\theta}_{L\ \boldsymbol{y}(i)}^W / \nabla\boldsymbol{\theta}_{L\,\boldsymbol{y}(i)}^b \tag{1}$$

where $i$ is the index of the sample within the input batch and $i \in [1, B]$. After disaggregating the feature map of each data individually, the private data can be generated quickly by inputting the disaggregated feature map into the generator (Xue et al., 2023), which is trained by inputting the feature maps (generated by feeding the data from the auxiliary dataset into the target model) and generating the corresponding data from the auxiliary dataset. FGLA (Xue et al., 2023) can reconstruct private data quickly since it only requires analytical disaggregation and generation, thus simplifying the reconstruction data problem to a feature map extraction problem.

However, this analytical disaggregation in Eq. 1 may be ineffective for the data with the same labels. Actually, the feature map from the disaggregation in Eq. 1 is a linear combination of the feature maps of all the data individually entered into the model (Fowl et al., 2021), and the data with the same label will resolve the same feature map. For example, for data samples $\boldsymbol{x}_i$ and $\boldsymbol{x}_j$ with the same label in batch $\boldsymbol{x}$, we can have $\boldsymbol{y}(i) = \boldsymbol{y}(j)$, and thus we arrive at $\boldsymbol{f}_{\boldsymbol{y}(i)} = \boldsymbol{f}_{\boldsymbol{y}(j)}$ through Eq. 1. Therefore, we have to resolve to another feature map extraction method.

We observe that more previous work (Zhu & Blaschko, 2020) utilized Eq. 2 to analytically invert the cross-entropy loss function to obtain the model output $\hat{\boldsymbol{y}}$ for the input batch, and then derived the original private data recursively through the gradient. This method inspired the idea that we can disaggregate the feature map by first leaking the model output and then disaggregating it. However,

this method only works in a limited setting of a batch size of 1, and a number of classes of 2.

$$\nabla \boldsymbol{\theta}_L^W \cdot \boldsymbol{\theta}_L^W = \frac{\partial l(\hat{\boldsymbol{y}}, \boldsymbol{y})}{\partial \hat{\boldsymbol{y}}} \cdot \hat{\boldsymbol{y}} \tag{2}$$

## 4.2 DATA LEAKAGE FROM GRADIENTS

Nevertheless, such fast data reconstruction by disaggregating the feature maps still attracts us to explore the feature map disaggregation methods in a more generic FL setting. To this end, we propose three innovative equations to disaggregate the model outputs and hence the feature maps, finally generating the original data, which overcomes the above problems while maintaining the superior reconstruction capabilities of the above methods. Algorithm 1 and Fig. 2 outline the flow of our algorithm.

Considering a generic FL setting of batch size $> 1$, and the number of classes $> 2$, we improve Eq. 2 as Proposition 1:

**Proposition 1.** $\nabla \boldsymbol{\theta}_L^W \cdot \boldsymbol{\theta}_L^W = \frac{\partial l(\hat{\boldsymbol{y}}, \boldsymbol{y})}{\partial \hat{\boldsymbol{y}}} \cdot (\hat{\boldsymbol{y}} - \boldsymbol{\theta}_L^b)$, where $\boldsymbol{\theta}_L^W$ indicates the weights of FCL, $\boldsymbol{\theta}_L^b$ indicates the bias of FCL, $\hat{\boldsymbol{y}}$ denotes the model output for private data, $\boldsymbol{y}$ denotes the ground-truth label, and $l(\cdot, \cdot)$ denotes the cross-entropy loss function.

*Proof.* See Appendix A.1 for a detailed proof. □

Since $\nabla \boldsymbol{\theta}_L^W$, $\boldsymbol{\theta}_L^W$, $\boldsymbol{\theta}_L^b$, and $\boldsymbol{y}$ are known to the attacker, we can obtain the model output $\hat{\boldsymbol{y}}$ through Proposition 1. It can be observed that Proposition 1 establishes $C \times C$ equations through the matrix $\nabla \boldsymbol{\theta}_L^W \cdot \boldsymbol{\theta}_L^W$ of size $C \times C$ on the left side of the Proposition 1. Then, to fully utilize the data information embedded in the gradient of the bias and establish more equations, we establish the second equation:

$$\nabla \boldsymbol{\theta}_L^b = \frac{\partial l(\hat{\boldsymbol{y}}, \boldsymbol{y})}{\partial \hat{\boldsymbol{y}}} \cdot \frac{\partial \hat{\boldsymbol{y}}_{(:,j)}}{\partial \boldsymbol{\theta}_{Lj}^b} = \frac{\partial l(\hat{\boldsymbol{y}}, \boldsymbol{y})}{\partial \hat{\boldsymbol{y}}} \cdot \mathbb{I} \tag{3}$$

where the $\mathbb{I} \in \mathbb{R}^{B \times 1}$ and $\mathbb{I}_i = 1, 1 < i < B$. Since the $\nabla \boldsymbol{\theta}_L^b$, and $\boldsymbol{y}$ is available to the attacker, we can obtain the model output through Eq. 3, which provides $C$ equations. Given there are no appropriate mathematical techniques to analytically solve for model output $\hat{\boldsymbol{y}}$ directly through the system of equations provided by Proposition 1 and Eq. 3, we choose to employ an optimization technique to search for model output $\hat{\boldsymbol{y}}$. First, we initialize the $\hat{\boldsymbol{y}}$ randomly, which we call it dummy model output $\hat{\boldsymbol{y}}'$ and then continuously adjust and optimize it so that it satisfies the equations established in Proposition 1 and Eq. 3. Accordingly, we establish the following objective:

$$\mathcal{L}_1(\hat{\boldsymbol{y}}') = \alpha_1 \overbrace{\left\| \nabla \boldsymbol{\theta}_L^W \cdot \boldsymbol{\theta}_L^W - \frac{\partial l(\hat{\boldsymbol{y}}', \boldsymbol{y})}{\partial \hat{\boldsymbol{y}}'} \cdot (\hat{\boldsymbol{y}}' - \boldsymbol{\theta}_L^b) \right\|^2}^{Proposition 1: \mathcal{L}_w} + \alpha_2 \overbrace{\left\| \nabla \boldsymbol{\theta}_L^b - \frac{\partial l(\hat{\boldsymbol{y}}', \boldsymbol{y})}{\partial \hat{\boldsymbol{y}}'} \cdot \mathbb{I} \right\|^2}^{Eq.3: \mathcal{L}_b} \tag{4}$$

After obtaining a dummy model output $\hat{\boldsymbol{y}}'$, we can analytically disaggregate the feature map input into the last FCL through the right part of the Eq. 5:

$$\nabla \boldsymbol{\theta}_L^W = \frac{\partial l(\hat{\boldsymbol{y}}, \boldsymbol{y})}{\partial \hat{\boldsymbol{y}}} \cdot \frac{\partial \hat{\boldsymbol{y}}}{\partial \boldsymbol{\theta}_L^W} = \frac{\partial l(\hat{\boldsymbol{y}}, \boldsymbol{y})}{\partial \hat{\boldsymbol{y}}} \cdot \boldsymbol{f} \implies \boldsymbol{f}' = \left( \frac{\partial l(\hat{\boldsymbol{y}}', \boldsymbol{y})}{\partial \hat{\boldsymbol{y}}'} \right)^{-1} \cdot \nabla \boldsymbol{\theta}_L^W \tag{5}$$

Since $\hat{\boldsymbol{y}}'$ can be obtained by optimizing with the objective in Eq. 4, and $\boldsymbol{y}$ and $\nabla \boldsymbol{\theta}_L^W$ is available to the attacker, we can compute a dummy feature map $\boldsymbol{f}'$ directly through Eq. 5. Considering that matrix $\partial l(\hat{\boldsymbol{y}}', \boldsymbol{y})/\partial \hat{\boldsymbol{y}}'$ may not have a corresponding inverse matrix, we employ the "Moore-Penrose inverse" (Moore, 1920; Penrose, 1955) method to obtain its pseudo-inverse as it is the most widely known generalization of the inverse matrix (Ben-Israel & Greville, 2003; Campbell & Meyer, 2009). After obtaining the feature map, we feed it into the generator to invert the feature map to input space, and the training details of the generator are described in Subsection 4.1. Notably, the optimization results of the dummy model output $\hat{\boldsymbol{y}}'$ will directly affect the reconstruction results as the Eq. 5 for feature map computation is almost lossless.

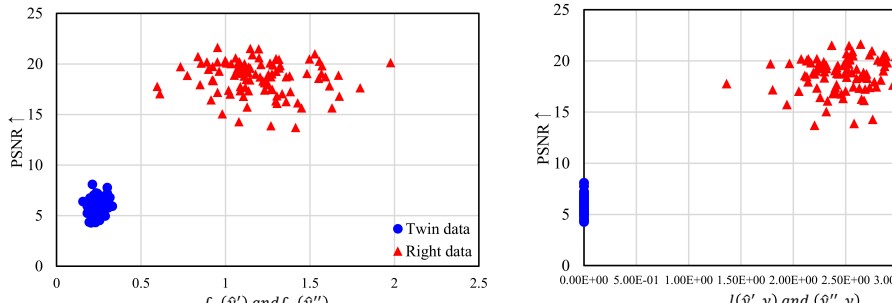

Figure 3: **Left**: PSNR values vs $\mathcal{L}_1(\hat{\boldsymbol{y}}')$ for right data and PSNR values vs $\mathcal{L}_1(\hat{\boldsymbol{y}}'')$ for twin data. **Right**: PSNR vs $l(\hat{\boldsymbol{y}}', \boldsymbol{y})$ for right data and PSNR vs $l(\hat{\boldsymbol{y}}'', \boldsymbol{y})$ for twin data.

However, model output $\hat{\boldsymbol{y}}$ is not always perfectly recovered through the optimization objective in Eq. 4. Proposition 1 and Eq. 3 provide a total of $C \times C + C$ equations, and the number of variables we need to search for the model output $\hat{\boldsymbol{y}}$ is $B \times C$. Thus the client's batch size $B$ directly determines the feasibility of recovering $\hat{\boldsymbol{y}}$ through the above equations as the number of classes $C$ for a selected target model is given. When $B > C + 1$, the established system of equations is underdetermined with the number of variables greater than the number of equations, thus we cannot recover unique $\hat{\boldsymbol{y}}$. We include increasing $B$ to greater than $C + 1$ into the limitation of our approach, as detailed in Section 6. When $B \leq C$, the established system of equations allows for the recovery of unique $\hat{\boldsymbol{y}}$. Unfortunately, there may be multiple solutions in the situation of $B \leq C$, of which only one is the right solution, and the remaining solutions are referred to in prior work R-GAP (Zhu & Blaschko, 2020) as the twin solutions due to the non-monotonicity of $\partial l(\hat{\boldsymbol{y}}, \boldsymbol{y})/\partial \hat{\boldsymbol{y}}$ (Zhu & Blaschko, 2020). Therefore, when $B <= C$, we need to observe the difference between the right solution and the twin solutions and use this difference to guide the optimization toward the right solution.

To find out the difference between the right data with higher PSNR values (a metric assessing the similarity between reconstructed and actual data) and the twin data with lower PSNR values, we conduct experiments and collect 100 batches of right data and 100 batches of twin data. The left of Fig. 3 shows their PSNR values vs $\mathcal{L}_1$. Unexpectedly, the twin data instead had lower $\mathcal{L}_1$ values. Considering that the reconstruction results depend only on the reconstructed dummy model outputs, our intuition suggests that the reconstructed dummy model outputs $\hat{\boldsymbol{y}}''$ of twin data are not close to the actual model outputs but are close to the ground-truth labels $\boldsymbol{y}$, which results in a much smaller $l(\hat{\boldsymbol{y}}'', \boldsymbol{y})$ of twin data than that of right data. So the gradient of the twin data is much smaller than that of the right data. Thus the gradient as well as $\partial l(\hat{\boldsymbol{y}}, \boldsymbol{y})/\partial \hat{\boldsymbol{y}}$ involved in $\mathcal{L}_1$ computation of the twin data is smaller, thus allowing the twin data to achieve a smaller $\mathcal{L}_1$ even without matching with the actual data. To verify our intuition, we show $l(\hat{\boldsymbol{y}}', \boldsymbol{y})$ for 100 batches right data and $l(\hat{\boldsymbol{y}}'', \boldsymbol{y})$ for 100 batches twin data in the right of Fig. 3, which proves our intuition. Therefore, we can guide the optimization towards the right data by constraining the cross-entropy loss between the dummy model output $\hat{\boldsymbol{y}}'$ and the ground-truth label $\boldsymbol{y}$. We have the following Proposition for approximating the cross-entropy loss of the attacked batch to guide the optimization of the dummy model output:

**Proposition 2.** $l(\hat{\boldsymbol{y}}, \boldsymbol{y}) \approx -\frac{1}{B} \sum_{i=1}^{B} \ln \left( \nabla \boldsymbol{\theta}^b_{L\boldsymbol{y}_{(i)}} + \boldsymbol{\lambda}_{y_{(i)}}/B \right)$. *Where $\boldsymbol{y}$ indicates the ground-truth label, $\boldsymbol{y}(i)$ indicates the ground-truth label for the $i^{th}$ sample in a batch, $\nabla \boldsymbol{\theta}^b_L$ represents the gradient of the bias term in the FCL, and $\boldsymbol{\lambda}_j$ indicates the occurrence number of label $j$ in the $\boldsymbol{y}$.*

*Proof.* See Appendix A.2 for a detailed proof. □

Appendix A.3 provides an approximation error analysis and discussion about when and why approximations in Proposition 2 hold. Proposition 2 regularizes the established system of equations, guiding the optimization towards the right solution, even though it only offers one equation. Based on Proposition 2, we can have a new optimization objective:

$$\mathcal{L}_{loss}(\hat{\boldsymbol{y}}') = \left\| l(\hat{\boldsymbol{y}}', \boldsymbol{y}) - [-\frac{1}{B} \sum_{i=1}^{B} \ln(\nabla \boldsymbol{\theta}^b_{L\boldsymbol{y}(i)} + \frac{\boldsymbol{\lambda}_{\boldsymbol{y}(i)}}{B})] \right\|^2 \quad (6)$$

Therefore, the total optimization objective for optimizing the dummy model output is:

$$\mathcal{L}_{total}(\hat{\boldsymbol{y}}') = \alpha_1 \mathcal{L}_w(\hat{\boldsymbol{y}}') + \alpha_2 \mathcal{L}_b(\hat{\boldsymbol{y}}') + \alpha_3 \mathcal{L}_{loss}(\hat{\boldsymbol{y}}') \quad (7)$$

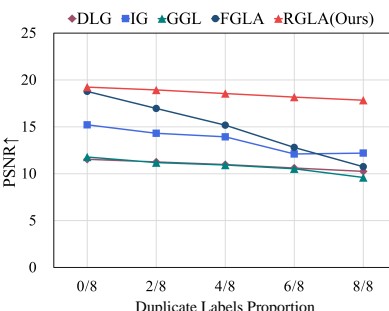 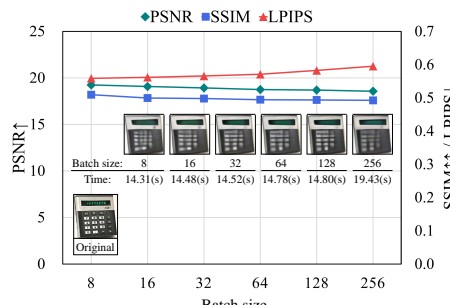

Figure 4: **Left**: When batch size is 8, the effect of the number of data sharing duplicate labels in a batch on the attack methods. **Right**: The curve of the effect of batch size on reconstruction result. RGLA is able to reconstruct the original data even when the batch size is 256.

Appendix A.4 shows that the model output reconstructed through the total optimization objective is close to the actual model output. Compared to optimization-based methods, our method searches the model output ($\mathcal{O}(B \cdot C)$), which is independent of the resolution size of the input data and has far fewer variables than the number of variables in the model input ($\mathcal{O}(B \cdot Channel \cdot Height \cdot Width)$) and not sensitive to initialization (refer to Appendix A.5). Moreover, our method is not affected by the duplicate label, whereas analytics-based methods cannot reconstruct data with duplicate labels.

## 5 EXPERIMENTS

**Setup.** We evaluate our RGLA on a hardware platform equipped with a 3.9GHz CPU i9-12900K, 128GB RAM, an NVIDIA 3090 GPU, and a deep learning environment with Python 3.8, CUDA 11.6, cuDNN 8.3.02, and PyTorch 1.13.1. We choose four classic datasets as user training datasets: ImageNet (Deng et al., 2009), CIFAR-10, CIFAR-100 (Krizhevsky, 2009), and CelebA (Liu et al., 2014) and take ImageNet dataset as default auxiliary dataset. All images are scaled to $224 \times 224$ pixels. We use the pre-trained ResNet50 network provided by Pytorch as the default global model for image classification in FL. We employ the Adam optimizer (Kingma & Ba, 2014) with a learning rate of 0.001 to optimize the dummy output with a total iterations of 20000 and set $\alpha_1 = 10000$, $\alpha_2 = 1$, and $\alpha_3 = 100$, which were adjusted based on experiment experience. The generator model's structure can be found in Xue et al. (2023). We measure the similarity between the original and reconstructed data using three commonly-used metrics: peak signal-to-noise ratio (PSNR↑), structural similarity (SSIM↑), and perceptual image similarity score (LPIPS↓) (Zhang et al., 2018). In our experiment, the default batch size is 8, and the default label distribution within a batch is randomly assigned. This random selection ensures diversity as we sample the batch from the dataset.

### 5.1 COMPARISON WITH THE STATE-OF-THE-ART METHODS

We compare RGLA with the state-of-the-art methods including optimization-based methods, DLG (Zhu et al., 2019), IG (Geiping et al., 2020), Generative Gradient Leakage (GGL) (Li et al., 2022) and analytics-based attack FGLA (Xue et al., 2023) in terms of reconstructing batch data with increasing proportions of duplicate labels. To ensure fairness in comparison, RGLA and all other state-of-the-art methods involved in the comparison use ground-truth labels for their attacks. The left figure of Fig. 4 illustrates that the performance of other methods deteriorates as the proportion of duplicate label data in the batch increases. When the proportion of data with duplicate labels is high, these methods fail to recover private data. The possible reason for this phenomenon is that FGLA cannot reconstruct data with duplicate labels, as described in Section 4. Moreover, the entanglement in the gradient rows of the data with the same labels complicates the optimization process of DLG, IG, and GGL, leading to a degradation in the quality of the reconstructed results. This phenomenon is also described in the prior work (Qian & Hansen, 2020). In contrast, RGLA demonstrates robust performance even with a high proportion of data containing duplicate labels. We provide the tabular form of the left figure in Fig. 4 and the visual comparison results on one batch in Appendix A.6.

### 5.2 EVALUATION ON INCREASING BATCH SIZES AND DIFFERENT DATASETS

To assess the effect of batch size on RGLA, we conduct experiments with increasing batch size from 8 to 256. The reported results averaged on the outcomes of 100 batches in Fig. 4 reveal a gradual

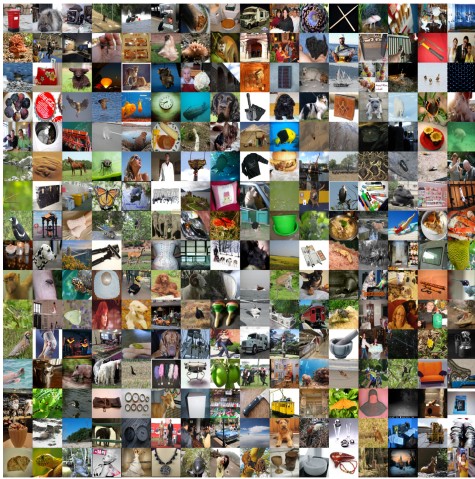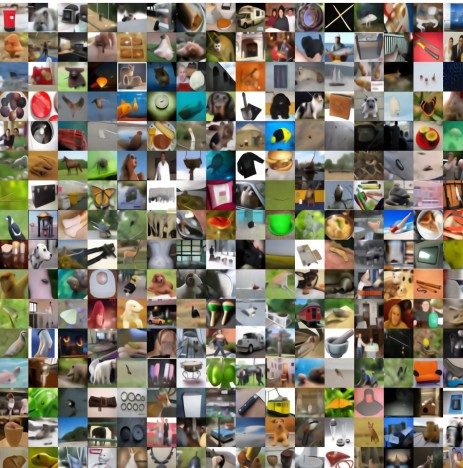

Figure 5: **Left:** Ground truth batch of 256 client images. **Right:** visual reconstructed results.

decline in performance as the batch size increases. This decline can be attributed to the increased complexity resulting from a greater number of variables in the model output. The visual example illustrated in Fig. 5 demonstrates that our method can still work even with a batch size of 256. Given that FGLA (Xue et al., 2023) presents an outperformance over other state-of-the-art methods with increasing batch size and other state-of-the-art methods perform poorly with a batch size of 8 as present in the left of Fig. 4, we compare the performance of RGLA with the performance of FGLA with the increasing batch size and the experiment results can be found in Appendix A.8.

In order to verify the generalizability of our method, we validate our RGLA on the CelebA dataset (face classification task with 10177 categories) and the CIFAR-100 dataset. The right of Fig. 2 presents the experimental results on CIFAR-100 and CelebA datasets, including the metrics averaged over 100 data batches and a visual example. These results effectively showcase the outstanding performance of our method across diverse datasets, emphasizing the robustness and generalizability of RGLA. Despite the blurry reconstructed results on the CelebA facial dataset, there are still some attributes of the attacked images exposed, such as background, hair color, skin tone, and facial contours. It's essential to note that in the real-world FL system, the adversary might not always have precise knowledge about the client's dataset. Therefore, it is important for GLA attacks to be robust when the auxiliary dataset is different from the target dataset.

## 5.3 EVALUATION AGAINST POSSIBLE DEFENSE STRATEGIES

We assess the effectiveness of our RGLA and other state-of-the-art methods against three relatively strict defense mechanisms, similar to the ones used in the prior study (Li et al., 2022), which include (1) Noisy gradient, where Gaussian noise $\mathcal{N}(0, \sigma^2)$ is added to the gradients (Sun et al., 2020); (2) Gradient Clipping, where the gradient values are clipped with a bound $\mathcal{C}$ (Geyer et al., 2017); and (3) Gradient Compression, where gradients with smaller magnitudes are set to zero (Zhu et al., 2019). Specifically, we add noise to the gradient with $\sigma = 0.01$ (Sun et al., 2020), clip the gradients with $\mathcal{C} = 4$ (Abadi et al., 2016), and compress the gradient (Zhu et al., 2019) with a compression ratio of 99.9%. Table 2 compares the performance of our proposed RGLA method and the state-of-the-art GLA methods under these strict defense settings. Our method performs well even under these three strict defense mechanisms, particularly in defending against gradient compression, while gradient compression is frequently employed in FL systems to reduce bandwidth (Lin et al., 2017). This is

Table 2: The comparison of the performance of our RGLA and the state-of-the-art GLA methods under three relatively strict defense mechanisms.

| Method | Noisy gradient (Sun et al., 2020) | | | Gradient Clipping (Geyer et al., 2017) | | | Gradient Compression (Zhu et al., 2019) | | |
|---|---|---|---|---|---|---|---|---|---|
| | PSNR↑ | SSIM↑ | LPIPS↓ | PSNR↑ | SSIM↑ | LPIPS↓ | PSNR↑ | SSIM↑ | LPIPS↓ |
| DLG (Zhu et al., 2019) | 11.90780 | 0.09130 | 1.11990 | 9.94580 | 0.10090 | 1.17470 | 11.19410 | 0.07190 | 1.23530 |
| IG (Geiping et al., 2020) | 14.25990 | 0.44350 | 0.57790 | 12.55480 | 0.42270 | **0.54570** | 12.45969 | 0.44249 | 0.80062 |
| GGL (Li et al., 2022) | 11.26461 | 0.32251 | 0.56769 | 10.19317 | 0.27268 | 0.56417 | 10.55477 | 0.27966 | 0.60754 |
| FGLA (Xue et al., 2023) | 18.84118 | 0.48547 | 0.55527 | **17.43373** | **0.46835** | 0.63645 | 10.45432 | 0.26430 | 0.88357 |
| RGLA(Ours) | **18.90560** | **0.52940** | **0.51770** | 16.13298 | 0.41319 | 0.68246 | **16.28875** | **0.48409** | **0.60309** |

| Optimization objectives | PSNR↑ | SSIM↑ | LPIPS↓ |
|---|---|---|---|
| $\mathcal{L}_w$ | 14.55934 | 0.38931 | 0.71797 |
| $\mathcal{L}_w+\mathcal{L}_{loss}$ | 19.16308 | 0.50066 | 0.57047 |
| $\mathcal{L}_w+\mathcal{L}_b$ | 14.55934 | 0.38931 | 0.71797 |
| $\mathcal{L}_w+\mathcal{L}_b+\mathcal{L}_{loss}$ | **19.24153** | **0.50948** | **0.55870** |

Table 3: Comparison of average outcomes across 100 batches for different optimization objectives combinations.

| Batch size | PSNR↑ | SSIM↑ | LPIPS↓ |
|---|---|---|---|
| B=64 | 20.67648 | 0.79191 | 0.328131 |
| B=128 | 7.852459 | 0.316369 | 0.929359 |

Table 4: Comparison of reconstruction quality metrics for two scenarios $B \times C < C \times (C + 1) + 1$ ($B = 64, C = 100$) and $B \times C > C \times (C + 1) + 1$ ($B = 128, C = 100$).

because all three defense methods only produce changes to some parts of the gradient. For example, additive Gaussian noise adds Gaussian noise 0 to a portion of the gradient, gradient clipping retains gradients below a certain bound, and gradient compression retains gradients with large absolute values. RGLA leverages these unchanged gradients to leak the model output and further leak the original private data. One unexpected finding is that adding noise to the gradient and clipping the gradient had little effect on the FGLA method. This is because the clipping defense is an operation on a gradient: $\nabla \boldsymbol{\theta}' \leftarrow \frac{\mathcal{C}}{\|\nabla \boldsymbol{\theta}\|} \cdot \nabla \boldsymbol{\theta}$. The FGLA method extracts the feature map by $\boldsymbol{f}' \approx \frac{\nabla \boldsymbol{\theta}_L^{W'}}{\nabla \boldsymbol{\theta}_L^{b'}} = \frac{\nabla \boldsymbol{\theta}_L^W \cdot \frac{\mathcal{C}}{\|\nabla \boldsymbol{\theta}\|}}{\nabla \boldsymbol{\theta}_L^b \cdot \frac{\mathcal{C}}{\|\nabla \boldsymbol{\theta}\|}}$, thus eliminating the clipping defense. See Appendix A.10 for the visual reconstruction example of RGLA and other state-of-the-art methods under these three defense mechanisms.

## 5.4 ABLATION STUDY

Next, we perform ablation experiments to explore the functionality of the three optimization objectives. Given $\mathcal{L}_w$ provides the maximum number of equations $C \times C$, we regard $\mathcal{L}_w$ as paramount among the three optimization objective components. To assess the feasibility of relying solely on $\mathcal{L}_w$, and discern the auxiliary roles of $\mathcal{L}_b$ and $\mathcal{L}_{loss}$ for the optimization, we execute experiments with different optimization objective combinations as shown in Table 3, while keeping other settings consistent. A comparison of the first and fourth rows in Table 3 reveals that relying solely on $\mathcal{L}_w$ is insufficient for model output recovery. A comparative analysis between the first and third rows, and the second and fourth rows, suggests that $\mathcal{L}_b$ seemingly plays an insignificant role. This is because the dataset is ImageNet with $C = 1000$ thus the number of equations provided by $\mathcal{L}_w$ ($C \times C$) vastly overshadows that provided by $\mathcal{L}_b$ ($C$). However, when the magnitude of $C$ diminishes, $\mathcal{L}_b$ manifests its utility. Specifically, when the batch size increases to $C \times C + 1 < B \times C < C \times (C + 1) + 1$, $\mathcal{L}_b$ plays a decisive role in the success of reconstruction. Additionally, from Table 3, we can also see that $\mathcal{L}_{loss}$ regularizes $\mathcal{L}_w$ and $\mathcal{L}_b$, meaning that $\mathcal{L}_{loss}$ distinguishes between right data and twin data, guiding the optimization towards the right solution.

## 6 CONCLUSION AND FUTURE WORK

Our RGLA tackles the challenge of high-resolution data and duplicate labels in GLAs, expanding the boundaries of gradient leakage attacks. Through optimizing the model output of the batch, RGLA successfully disaggregates the original model output of each sample within a batch from the averaged gradient, enabling disaggregation of each original input by utilizing the isolated original model output. Extensive experiments demonstrate the outstanding performance of RGLA compared to existing state-of-the-art methods, exhibiting robustness to label distribution, batch size, possible defense mechanisms, and initialization methods. Finally, we emphasize the importance of continuously exploring and improving privacy-preserving mechanisms in distributed machine learning.

**Limitation.** The proposed method does not work when the batch size $B$ is larger than $C + 1$ since there are more the variables of model output ($B \times C$) than the number of equations ($C \times (C+1)+1$) provided by our optimization objectives. To show this limitation, we evaluate our method in the following two scenarios: 1) $B \times C < C \times (C + 1) + 1$ and 2) $B \times C > C \times (C + 1) + 1$. To this end, we take the CIFAR-100 with the number of classes $C$ of 100 and successively set the batch size to 64 and 128. From the experiment results in Table 4, we can see that our method performs well when $B \times C < C \times (C + 1) + 1$, but it performs poorly when the number of variables of model output is more than $C \times (C + 1) + 1$. We hope that future research will be able to successfully extend this effective attack to scenarios where $B > C + 1$.

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

# A APPENDIX

## A.1 PROOF OF PROPOSITION 1

**Proposition.** $\nabla\boldsymbol{\theta}_L^W \cdot \boldsymbol{\theta}_L^W = \dfrac{\partial l(\hat{\boldsymbol{y}}, \boldsymbol{y})}{\partial \hat{\boldsymbol{y}}} \cdot (\hat{\boldsymbol{y}} - \boldsymbol{\theta}_L^b)$, where $\boldsymbol{\theta}_L^W$ indicates the weights of FCL, $\boldsymbol{\theta}_L^b$ indicates the bias of FCL, $\hat{\boldsymbol{y}}$ denotes the model output for private data, $\boldsymbol{y}$ denotes the ground-truth label, and $l(\cdot, \cdot)$ denotes the cross-entropy loss function.

*Proof.* The forward propagation of FCL can be written as $\hat{\boldsymbol{y}} = \boldsymbol{f} \cdot \boldsymbol{\theta}_L^W + \boldsymbol{\theta}_L^b$, thus we can have:

$$\frac{\partial \hat{\boldsymbol{y}}}{\partial \boldsymbol{\theta}_L^W} = \boldsymbol{f}, \tag{8}$$

based on the chain rule of derivation, we can have the following:

$$\nabla\boldsymbol{\theta}_L^W \cdot \boldsymbol{\theta}_L^W = \frac{\partial l(\hat{\boldsymbol{y}}, \boldsymbol{y})}{\partial \hat{\boldsymbol{y}}} \cdot \frac{\partial \hat{\boldsymbol{y}}}{\partial \boldsymbol{\theta}_L^W} \cdot \boldsymbol{\theta}_L^W, \tag{9}$$

we substitute $\dfrac{\partial \hat{\boldsymbol{y}}}{\partial \boldsymbol{\theta}_L^W}$ in Eq. (9) with its expression from Eq. (8) as follows:

$$\nabla\boldsymbol{\theta}_L^W \cdot \boldsymbol{\theta}_L^W = \frac{\partial l(\hat{\boldsymbol{y}}, \boldsymbol{y})}{\partial \hat{\boldsymbol{y}}} \cdot \frac{\partial \hat{\boldsymbol{y}}}{\partial \boldsymbol{\theta}_L^W} \cdot \boldsymbol{\theta}_L^W = \frac{\partial l(\hat{\boldsymbol{y}}, \boldsymbol{y})}{\partial \hat{\boldsymbol{y}}} \cdot \boldsymbol{f} \cdot \boldsymbol{\theta}_L^W = \frac{\partial l(\hat{\boldsymbol{y}}, \boldsymbol{y})}{\partial \hat{\boldsymbol{y}}} \cdot (\hat{\boldsymbol{y}} - \boldsymbol{\theta}_L^b) \tag{10}$$

$\square$

## A.2 PROOF OF PROPOSITION 2

**Proposition.** $l(\hat{\boldsymbol{y}}, \boldsymbol{y}) \approx -\frac{1}{B}\sum_{i=1}^{B} \ln\left(\nabla\boldsymbol{\theta}_{L\,\boldsymbol{y}(i)}^b + \boldsymbol{\lambda}_{y(i)}/B\right)$. Where $\boldsymbol{y}$ indicates the ground-truth label, $\boldsymbol{y}(i)$ indicates the ground-truth label for the $i^{th}$ sample in a batch, $\nabla\boldsymbol{\theta}_L^b$ represents the gradient of the bias term in the FCL, and $\boldsymbol{\lambda}_j$ indicates the occurrence number of label $j$ in the $\boldsymbol{y}$.

*Proof.* In a classification task with a cross-entropy loss function, the loss function can be defined as Eq.11 (Wainakh et al., 2021):

$$l(\hat{\boldsymbol{y}}, \boldsymbol{y}) = -\frac{1}{B}\sum_{i=1}^{B} \ln \frac{e^{\hat{\boldsymbol{y}}_{(i,\boldsymbol{y}(i))}}}{\sum_{j=1}^{C} e^{\hat{\boldsymbol{y}}_{(i,j)}}}, \tag{11}$$

where $\hat{\boldsymbol{y}}$ indicates the model output, which is also the output of the FCL and $\hat{\boldsymbol{y}}_{(i,j)}$ indicates the logit value for $j^{th}$ class when $i^{th}$ sample in a batch is given as input to the model. Based on the prior work (Wainakh et al., 2021), we can write the gradient value $\nabla\boldsymbol{\theta}_{Lj}^b$ w.r.t. the bias connected to the $j^{th}$ output representing the $j^{th}$ class confidence in the output layer as following Eq.12:

$$\nabla\boldsymbol{\theta}_{Lj}^b = \nabla\hat{\boldsymbol{y}}_{(:,j)} \cdot \frac{\partial \hat{\boldsymbol{y}}_{(:,j)}}{\partial \boldsymbol{\theta}_{Lj}^b} = \nabla\hat{\boldsymbol{y}}_{(:,j)} \cdot \mathbb{I} = -\frac{\boldsymbol{\lambda}_j}{B} + \frac{1}{B}\sum_{i=1}^{B} \frac{e^{\hat{\boldsymbol{y}}_{(i,\boldsymbol{y}(i))}}}{\sum_{t=1}^{C} e^{\hat{\boldsymbol{y}}_{(i,t)}}}, \tag{12}$$

when the $j$ in Eq. (12) equal to $\boldsymbol{y}(k^*)$, then we have Eq.13 (Wainakh et al., 2021):

$$\nabla\boldsymbol{\theta}_{L\,\boldsymbol{y}(k^*)}^b = -\frac{\boldsymbol{\lambda}_{\boldsymbol{y}(k^*)}}{B} + \frac{1}{B}\sum_{i=1}^{B} \frac{e^{\hat{\boldsymbol{y}}_{(i,\boldsymbol{y}(i^*))}}}{\sum_{j=1}^{C} e^{\hat{\boldsymbol{y}}_{(i,j)}}}, \tag{13}$$

where $k^* \in [1, B]$ denotes the index of a specific sample within a batch. In the initialization or pre-training phase of model training, all logit values in the model output $\hat{\boldsymbol{y}}$ are similar, due to the model's lack of discriminative ability over the input samples. This phenomenon can be expressed as:

$$\hat{\boldsymbol{y}}(i, j) \approx \hat{\boldsymbol{y}}(k, t), \tag{14}$$

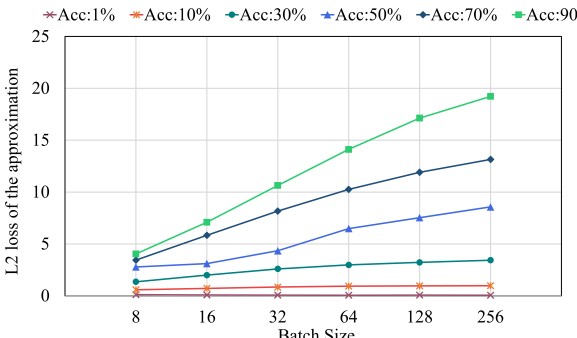

Figure 6: The curve of the effect of the accuracy of the model and batch size on the $L2$ error of the approximation

where $i, k \in [1, B]$ indicate the index of the sample within the input batch, and $j, t \in [1, C]$ indicate the index of the classes. Based on this observation and the Eq.13, we can further formulate the following equation:

$$\frac{e^{\hat{\boldsymbol{y}}_{(k^*, \boldsymbol{y}(k^*))}}}{\sum_{j=1}^{C} e^{\hat{\boldsymbol{y}}_{(k^*, j)}}} \approx \frac{1}{B} \sum_{i=1}^{B} \frac{e^{\hat{\boldsymbol{y}}_{(i, \boldsymbol{y}(k^*))}}}{\sum_{j=1}^{C} e^{\hat{\boldsymbol{y}}_{(i, j)}}} = \nabla \boldsymbol{\theta}^b_{L \boldsymbol{y}(k^*)} + \frac{\boldsymbol{\lambda}_{\boldsymbol{y}(k^*)}}{B}, \tag{15}$$

Then, we replace $\frac{e^{\hat{\boldsymbol{y}}_{(i, \boldsymbol{y}(i))}}}{\sum_{j=1}^{C} e^{\hat{\boldsymbol{y}}_{(i, j)}}}$ in Eq. (11) with the expression from Eq. 15, as follows:

$$l(\hat{\boldsymbol{y}}, \boldsymbol{y}) = -\frac{1}{B} \sum_{i=1}^{B} \ln \frac{e^{\hat{\boldsymbol{y}}_{(i, \boldsymbol{y}(i))}}}{\sum_{j=1}^{C} e^{\hat{\boldsymbol{y}}_{(i, j)}}} \approx -\frac{1}{B} \sum_{i=1}^{B} \ln \left( \nabla \boldsymbol{\theta}^b_{L \boldsymbol{y}(i)} + \frac{\boldsymbol{\lambda}_{\boldsymbol{y}(i)}}{B} \right) \tag{16}$$

$\square$

### A.3 Approximation Error Analysis

From the theoretical proof in A.2 of the approximation in Proposition 2, we learn that the validity of the approximation relies heavily on the similarity of the logit values in the model output of the low-accuracy model as stated in Eq. 14. This raises a number of questions: how does the accuracy of the model relate to the validity of the approximation in Proposition 2? At what point does this approximation not hold?

Table 5: Average $L2$ errors of approximation at the models with different accuracies and different batch sizes

| Batch size | 8 | 16 | 32 | 64 | 128 | 256 |
|---|---|---|---|---|---|---|
| Model with 1% accuracy | 0.1044 | 0.0826 | 0.0677 | 0.0615 | 0.0663 | 0.0594 |
| Model with 10% accuracy | 0.5764 | 0.7175 | 0.8448 | 0.9321 | 0.9553 | 0.9680 |
| Model with 30% accuracy | 1.3572 | 1.9977 | 2.5959 | 2.9827 | 3.2270 | 3.4328 |
| Model with 50% accuracy | 2.7877 | 3.1090 | 4.3454 | 6.4839 | 7.5381 | 8.5707 |
| Model with 70% accuracy | 3.4399 | 5.8314 | 8.1713 | 10.2586 | 11.9053 | 13.1496 |
| Model with 90% accuracy | 4.0387 | 7.0818 | 10.6421 | 14.1113 | 17.1310 | 19.2156 |

To answer these questions, we first train the models on CIFAR-100 with accuracies of 1%, 10%, 30%, 50%, 70%, and 90%. Then, we conduct experiments with data batch of batch sizes 8, 16, 32, 64, 128, and 256 and compute the average approximation $L2$ errors over 100 batches. The experiment results in Table 5 and Fig. 6 show the approximation $L2$ error is small, confirming that the approximation in our Proposition 2 is reliable. From Fig. 6, we also observe that the approximation error increases with batch size, which is because larger batch sizes make it more difficult to estimate the $L2$ loss of an attacked batch. In addition, the increase in model accuracy also results in an increase in approximation error because higher accuracy models are better able to distinguish between different samples in a batch, resulting in dissimilar logit values in the corresponding model outputs, which increases the error in Eq. 14 and further amplifies the final approximation error. Therefore, our attack primarily occurs during the initialization or pre-training phase during the model training

because, during these stages, the model exhibits lower accuracy and weaker sample discrimination capabilities, resulting in a smaller approximation error. It needs to be emphasized that regardless of the stage at which the attack occurs, as long as it can successfully leak data, it is something we need to pay attention to and guard against.

## A.4 Evaluation of the ability to leak the model output

As discussed and analyzed in the previous Section 4, inverting the cross-entropy loss function for model output disaggregation is the most important part of our method. To assess the difference between the optimized model output and the actual model output, we conduct experiments and compute the $L_2$ distance between the dummy model outputs and the actual model outputs at batch sizes of 8, 16, 32, 64, 128, and 256. The experimental results in Table 6 show that our method can accurately disaggregate the model output. Fig. 7 shows that the image generated by the feature map derived from the dummy model output closely resembles the image generated by the feature map derived from the actual model output. This further demonstrates that our method is able to accurately capture the model output and accurately reconstruct the private data.

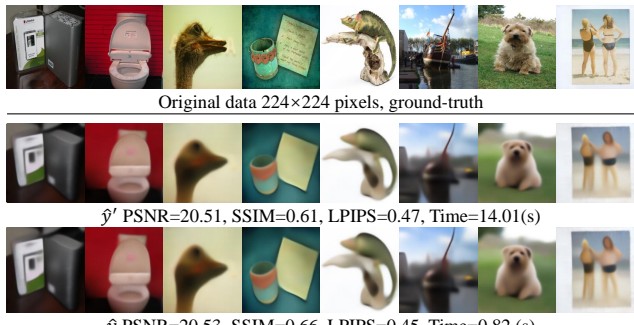

Original data 224×224 pixels, ground-truth

$\hat{y}'$ PSNR=20.51, SSIM=0.61, LPIPS=0.47, Time=14.01(s)

$\hat{y}$ PSNR=20.53, SSIM=0.66, LPIPS=0.45, Time=0.82 (s)

Figure 7: The comparison between the results reconstructed by the optimized model output $\hat{y}'$ and that by the actual model output $\hat{y}$ indicates the similarity between the reconstructed data of the optimized model output and that of the real model output. This similarity demonstrates the accuracy of our method in leaking the model output, and therefore, the model input.

**Remark 1.** *After the model output is exposed, any model inversion attack (Fredrikson et al., 2015; Wang et al., 2022; Nguyen et al., 2023) can be utilized to reveal the model input. Our proposed optimization method for the model output disaggregation has the ability to utilize all model inversion attacks to launch gradient leak attacks, thereby making the gradient more vulnerable.*

Table 6: The $L_2$ error between the optimized dummy model output and the actual model output at different batch sizes

| Batch size | 8 | 16 | 32 | 64 | 128 | 256 |
|---|---|---|---|---|---|---|
| $\|\hat{y}' - \hat{y}\|^2$ | 8.78E-05 | 0.0002 | 0.0007 | 0.0013 | 0.0032 | 0.0036 |

## A.5 Evaluation of the different initialization methods for the dummy model output

Existing optimization-based methods are often sensitive to dummy data initialization (Zhu & Blaschko, 2020; Yang et al., 2022), and inappropriate initialization may lead to the failure of these methods. So, a natural question arises: is our method also sensitive to the initialization of the dummy model outputs? To explore this, we employed six different initialization methods for the dummy model outputs: uniform distribution in the range [0, 1] (implemented using the torch.rand() function), standard normal distribution (implemented using torch.randn()), zero matrix initialization (implemented using torch.zeros()), one matrix initialization (implemented using torch.ones()), as well as initialization with model outputs for natural images and initialization with model outputs for the reconstruction results of FGLA (Xue et al., 2023). The experimental results in Table 7 clearly demonstrate that regardless of the initialization method used, our attack can successfully reconstruct the original data, and the results obtained from these different initialization methods are similar.

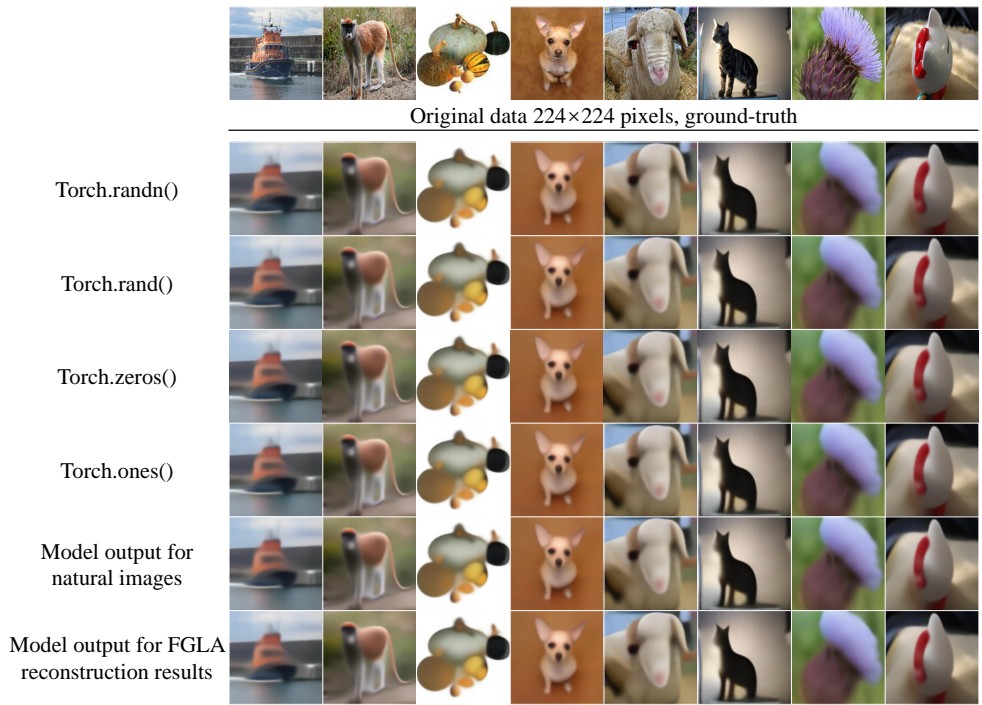

Figure 8: Visual examples of data reconstructions produced by various initialization methods. Our method consistently produces good reconstructions, showcasing its robustness to different initialization methods.

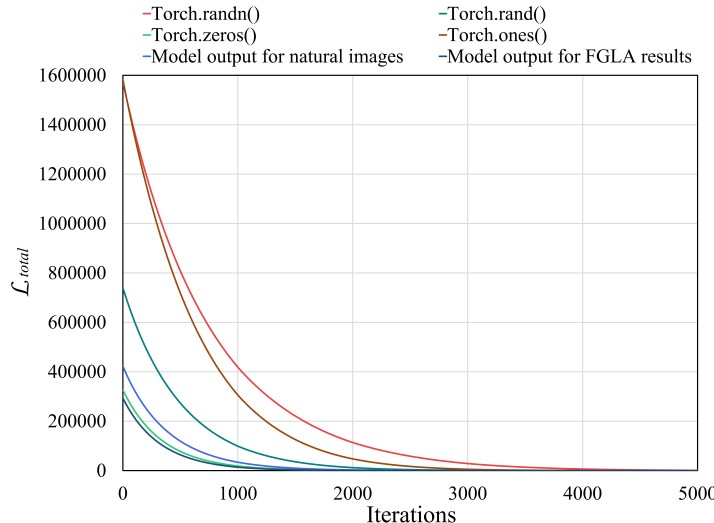

Figure 9: The effect of different initialization methods on the optimization convergence speed. Zero matrix initialization as well as using the model output for the reconstruction results of the FGLA as initialization requires fewer iterations to reach convergence.

This indicates that our method is not sensitive to initialization. Visual examples of one batch reconstructions using different initialization methods are shown in Fig. 8. Fig. 9 shows how the $\mathcal{L}_{total}$ varies with the number of iterations when optimizing using different initialization methods. It is clear from Fig. 9 that despite using different initialization methods, all of them eventually converge to the same minimum value. Moreover, using the zero matrix initialization or using the result of FGLA reconstruction as the initialization allows the optimization process to start closer to the final minimum point. This means that with these initialization methods, fewer iterations are required, which significantly reduces the time for the optimization to reach convergence.

Table 7: Comparison of our attack with different initialization methods for the dummy model outputs. The results demonstrate the insensitivity of our method to initialization.

| Initialization method | PSNR↑ | SSIM↑ | LPIPS↓ |
|---|---|---|---|
| Torch.randn() | 19.24153 | 0.50948 | 0.55870 |
| Torch.rand() | 19.24152 | 0.50948 | 0.55869 |
| Torch.zeros() | 19.24153 | 0.50948 | 0.55869 |
| Torch.ones() | 19.24153 | 0.50948 | 0.55869 |
| Model output for natural images | 19.24152 | 0.50948 | 0.55869 |
| Model output for FGLA reconstruction results | 19.24152 | 0.50948 | 0.55869 |

## A.6 COMPARISON WITH THE STATE-OF-THE-ART METHODS

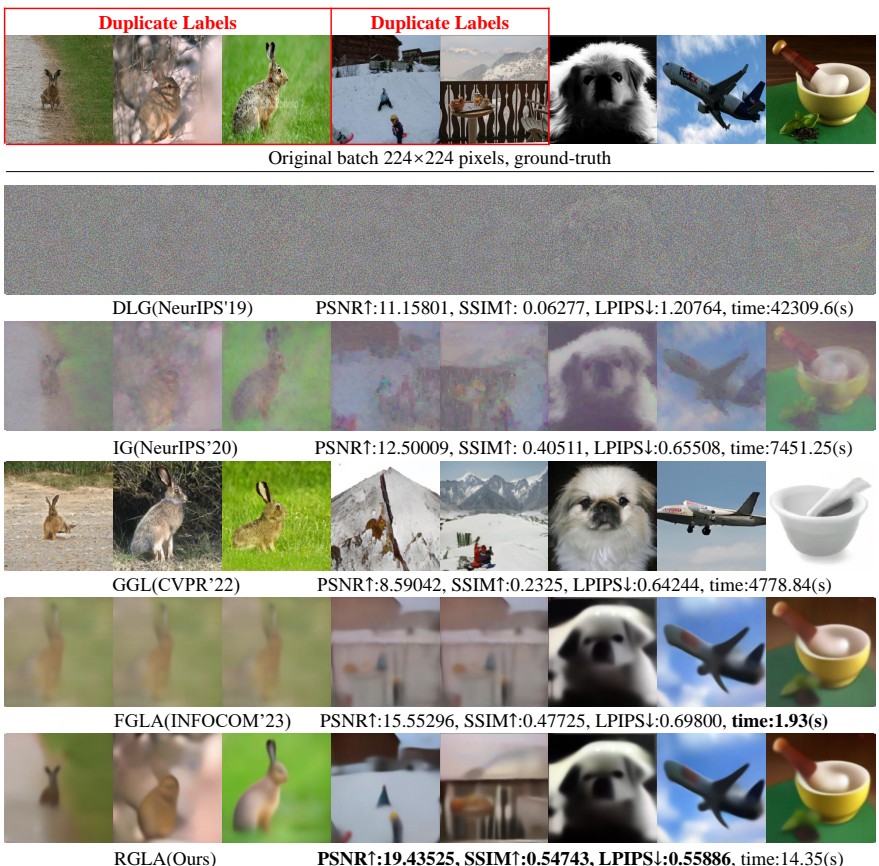

Figure 10: An example comparison of our method with the state-of-the-art GLA methods.

Table 8: When batch size is 8, the effect of the number of data sharing duplicate labels in a batch on the reconstruction results metric (PSNR↑) of the attack methods.

| Duplicate Labels Proportion | DLG | IG | GGL | FGLA | RGLA(Ours) |
|---|---|---|---|---|---|
| 0/8 | 11.53908 | 15.21782 | 11.78704 | 18.77932 | **19.24153** |
| 2/8 | 11.25802 | 14.31178 | 11.18857 | 16.96259 | **18.93582** |
| 4/8 | 10.97666 | 13.93340 | 10.93248 | 15.18163 | **18.55615** |
| 6/8 | 10.61047 | 12.10551 | 10.53121 | 12.80531 | **18.17758** |
| 8/8 | 10.24716 | 12.20067 | 9.59926 | 10.74843 | **17.84458** |

We provide the tabular form of the left of Fig. 4 in Table 8 and reconstruction example of RGLA and the stat-of-the-art GLAs in Fig. 10. As depicted in Fig. 10, existing optimization-based attacks face challenges in reconstructing high-resolution data, while the analysis-based attacks face challenges in reconstructing samples with duplicated labels within a batch. In contrast, RGLA effectively reconstructs data that closely resemble the original data in terms of evaluation metrics and

visual appearance. Among the existing methods, FGLA (Xue et al., 2023) demonstrates the fastest reconstruction time but fails to differentiate data with duplicate labels. The reason for this limitation is explained in Section 4. On the other hand, GGL (Li et al., 2022) generates high-quality data but struggles to accurately reconstruct the original data due to misclassification caused by duplicate labels. Moreover, optimization-based methods require at least 200 times more time to execute than our approach. Such methods optimize dummy inputs through gradient matching, which involves inputting dummy data into the model to obtain the dummy gradient and align it with the true gradient. That process is time-consuming.

### A.7 COMBINATION WITH PREVIOUS LABEL INFERENCE TECHNOLOGY

To better align our experiments with real-world scenarios, we combine our proposed RGLA attack with the state-of-the-art label inference technology (Ma et al., 2022), Instance-Level Reverse Gradient (iLRG) (Ma et al., 2022). We conducted experiments on CIFAR-100 and ResNet50 models and Table 9 shows the average results of data reconstruction on 100 batches. Notably, iLRG achieved a 100% accuracy rate in our experiments, though the order of the reconstructed labels differed from that of the actual ground-truth labels. As shown in Table 9, the reconstructed results obtained using ground-truth labels and those reconstructed results using the iLRG (Ma et al., 2022) are remarkably similar, albeit the label inference process required additional time. Fig. 11 provides visual reconstruction results, illustrating that both ground-truth labels and inferred labels can successfully reconstruct visual outcomes, except that using inferred labels causes the reconstructed data to be in a different order from the original data.

Table 9: Comparison of data reconstruction quality and time efficiency between ground-Truth label + RGLA and iLRG (Ma et al., 2022) + RGLA.

|  | PSNR↑ | SSIM↑ | LPIPS↓ | Time↓ |
|---|---|---|---|---|
| Ground-truth label + RGLA | 19.24153 | 0.50947 | 0.55870 | 14.35000 |
| iLRG (Ma et al., 2022) + RGLA | 19.22677 | 0.49973 | 0.56795 | 91.66502 |

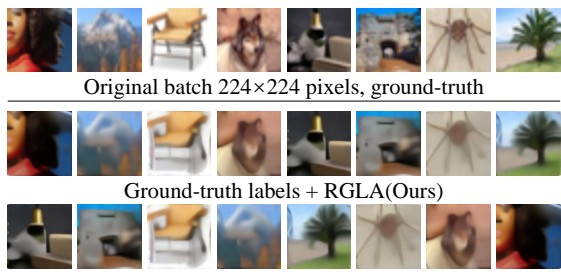

Original batch 224×224 pixels, ground-truth

Ground-truth labels + RGLA(Ours)

Inferred labels by iLRG + RGLA(Ours)

Figure 11: Visual reconstruction outcomes using ground-truth and inferred labels.

Table 10: Tabular form for the right figure in Fig. 4 and comparison with FGLA.

| Batch size | RGLA(Ours) | | | FGLA | | |
|---|---|---|---|---|---|---|
|  | PSNR↑ | SSIM↑ | LPIPS↓ | PSNR↑ | SSIM↑ | LPIPS↓ |
| 8 | 19.24153 | 0.50948 | 0.55870 | 18.43037 | 0.45977 | 0.60400 |
| 16 | 19.07956 | 0.49951 | 0.56184 | 18.42008 | 0.46021 | 0.61498 |
| 32 | 18.92351 | 0.49798 | 0.56576 | 18.41045 | 0.46034 | 0.66107 |
| 64 | 18.74563 | 0.49443 | 0.57093 | 18.33128 | 0.44717 | 0.69969 |
| 128 | 18.69326 | 0.49366 | 0.58253 | 18.17242 | 0.43765 | 0.71969 |
| 256 | 18.56861 | 0.49253 | 0.59527 | 18.08894 | 0.42596 | 0.74001 |

### A.8 EVALUATION ON INCREASING BATCH SIZES

We include the tabular form of the right figure of Fig. 4, along with a comparison to FGLA in Table 10. RGLA's attack performance slightly deteriorates and FGLA's attack performance worsens as the batch size increases. Our intuition for this phenomenon is that RGLA's method for disaggregating

the model output is optimization technology, and as the batch size increases, there are more variables to optimize, leading to poorer reconstruction. From Table 10, we can also observe that RGLA's performance against increasing batch sizes is superior to that of FGLA.

## A.9 EVALUATION ON HIGHER PIXELS RESOLUTIONS

Next, we evaluate the performance of the proposed attack on the image data of higher pixels 336×336. Apart from the difference in image resolution, all other experiment settings remain the same, including the target model of the ResNet50 model, the target dataset of the ImageNet dataset, and the batch size of 8. The experiment results in Table 11 show that our method is able to reconstruct the original private data regardless of whether the original data is under 224×224 pixels or 336×336 pixels. The decrease in the quality of the reconstruction results for the original data of 336×336 pixels can be attributed to the fact that the 336×336 pixels image is more complex and poses a greater challenge to the reconstruction capability of the generator. As a result, the intricate details of such complex images cannot be accurately captured during the reconstruction process, leading to overall quality degradation. Fig. 12 shows the visual reconstruction results of the same batch of images of different resolutions. These visualizations clearly show that our method maintains its effectiveness in reconstructing the visual appearance of an image despite the higher pixel sizes, demonstrating the robustness of our method at higher pixels.

Table 11: Performance metrics of RGLA on image data with resolutions of 224×224 and 336×336 pixels.

|  | PSNR↑ | SSIM↑ | LPIPS↓ | Time↓ |
|---|---|---|---|---|
| RGLA(224×224) | 18.80249 | 0.49666 | 0.58802 | 14.89383 |
| RGLA(336×336) | 16.75473 | 0.46615 | 0.38059 | 14.98800 |

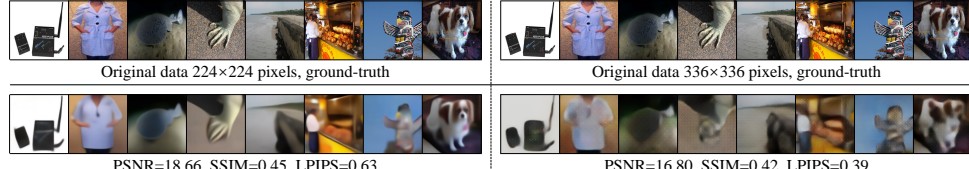

Figure 12: Visual reconstruction comparisons at different resolutions.

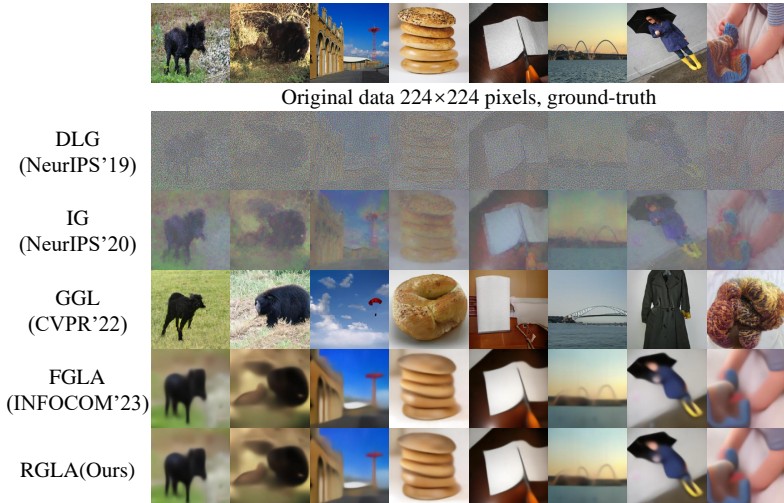

Figure 13: Comparison of the reconstruction results of RGLA and several state-of-the-art GLAs methods under privacy defense of additive noise $\sigma = 0.01$.

## A.10 EVALUATION AGAINST POSSIBLE DEFENSE STRATEGIES

This section contains the visual reconstruction comparison of RGLA and the state-of-the-art GLA attacks under noisy gradient, clipped gradient, and compressed gradient. From the visual compari-

son results presented in Fig. 13, Fig. 14, and Fig. 15, it is evident that adding noise to the gradients and clipping gradients has a negligible effect on FGLA and RGLA. However, gradient compression, which is a commonly used technique to reduce the bandwidth in the federated learning system, disables FGLA from recovering the original data. Conversely, RGLA can still reconstruct data that is similar to the original data even under gradient compression with a compression ratio of 99.9%. The data reconstructed by the GGL method is of high quality, owing to the bigger generator BigGAN (Brock et al., 2018), but it does not resemble the original data. Despite revealing some information about the original data, DLG and IG struggle to reconstruct it visually.

Given that FGLA has demonstrated a highly effective attack against compressed gradients compared to other state-of-the-art GLAs, we exclusively focus on conducting a comparative analysis between RGLA and FGLA under different levels of gradient compression. Table 12 shows the average PSNR values of the reconstructed results of the RGLA and FGLA methods on 100 batches as the gradient compression increases. From Table 12, we can see that both RGLA and FGLA remain equally effective until the gradient compression rate of 99.9%, but when the gradient compression reaches up to 99.9%, FGLA fails to reconstruct the original data, whereas RGLA retains the ability to reconstruct the original data. Fig. 16 provides the reconstruction results of the FGLA and RGLA attack visualizations as the compression rate increases.

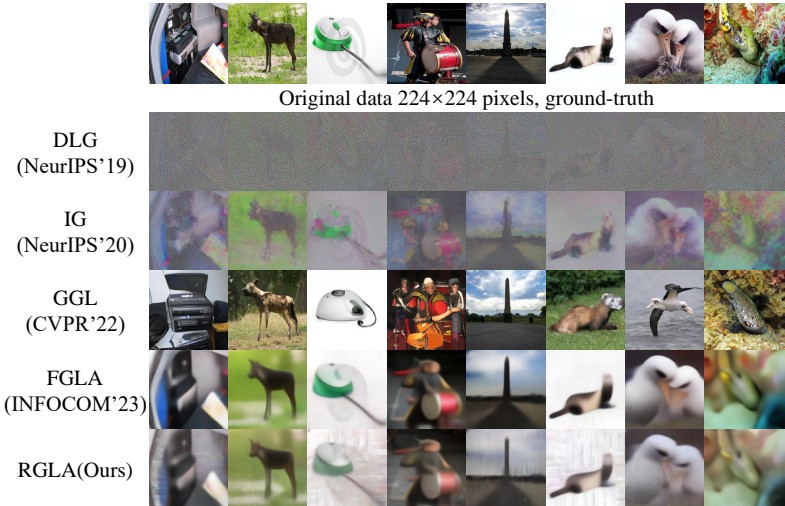

Figure 14: Comparison of the reconstruction results of RGLA and several state-of-the-art GLAs methods under privacy defense of clipping $C = 4$.

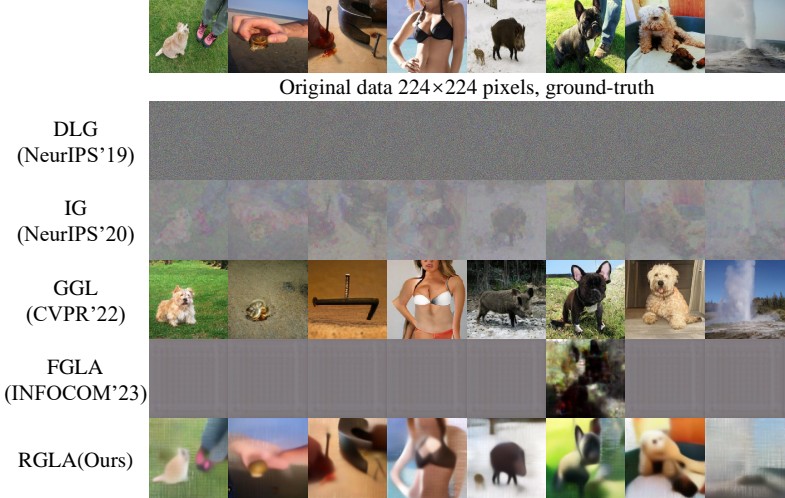

Figure 15: Comparison of the reconstruction results of RGLA and several state-of-the-art GLAs under gradient compression ratio 99.9%. Our RGLA was the only method whose reconstructed results resembled the original data even under gradient compression of compression ratio 99.9%.

Table 12: Average PSNR Values for RGLA and FGLA Methods under different gradient compression ratios.

|  | 0% | 80% | 90% | 99% | 99.90% |
|---|---|---|---|---|---|
| RGLA | 18.79353 | 18.79237 | 18.79329 | 18.79325 | 17.00678 |
| FGLA(Xue et al., 2023) | 18.77932 | 18.77535 | 18.77598 | 18.77587 | 10.43223 |

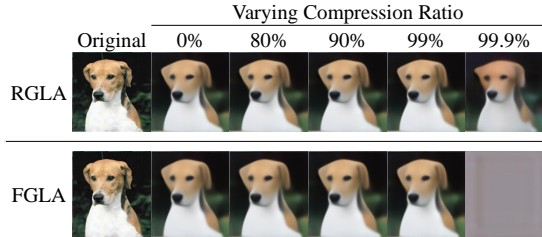

Figure 16: Visual reconstruction results of RGLA and FGLA under different gradient compression ratios.

## A.11 ABLATION STUDY

We provide a visual reconstruction example of different optimization objective combinations in Fig. 17.

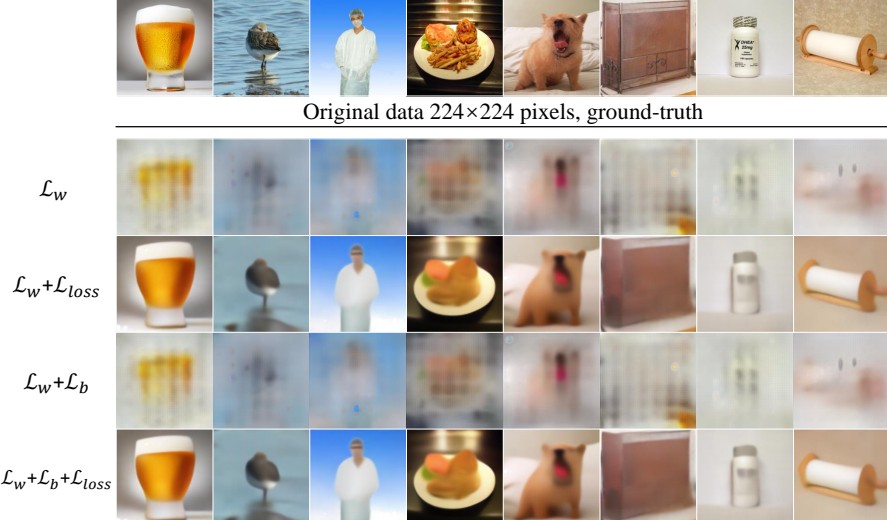

Figure 17: A visual reconstruction example of different optimization objective combinations.

