# OpenReview forum: "RGLA: Reverse Gradient Leakage Attack using Inverted Cross-Entropy Loss Function"
_ICLR.cc/2024/Conference — Submitted to ICLR 2024_

### Official Review · Reviewer_iafz · 2023-10-26

**Soundness:** 1 poor
**Presentation:** 1 poor
**Contribution:** 2 fair
**Rating:** 3
**Confidence:** 5

**Summary:**

The paper introduces a novel optimization-based data reconstruction attack on gradients of CNNs trained on classification tasks with cross-entropy losses based on two ingredients: 1. reconstructing the model outputs $\hat{y}$, which allows to recover the feature maps $\textbf{provided that one knows all labels}$ 2. inverting the feature map to retrieve the input using a known method. The authors also propose a simple defense mechanism against this attack based on increasing the batch-size.

**Strengths:**

- the paper introduces a novel loss and approach based on recovering $\hat{y}$
- the authors provide open-source code
- the authors' attack can target CNNs and not only fully connected neural networks although no pooling layer ca, be used
- the authors test the performance of their attack against different defense mechanisms based on clipping / compression / noise addition of the gradients and show robustness

**Weaknesses:**

Major.

- Few of the authors' claims are supported by evidence.
As an example let's take the sentence from the abstract"this is the $\textbf{first}$ successful disaggregation of the feature map in $\textbf{generic FL setting}$"

$\textbf{"first"}$.

It is not the first article to disaggregate feature maps, see for instance [1], important reference, $\textbf{which is missing in the paper}$.
Also much less important but very relevant work of [4] is missing as well. If the authors claim they target large batches of large resolutions, [4] would have been the reviewer's first choice as a baseline although [4] uses batch-norm statistics.

$\textbf{"generic FL setting"}$.

The attack setting described in the threat model by the authors is everything but generic because 1. contrary to most of its competition $\textbf{the attack requires access to the ground truth labels}$ (also the wording is weird "$\mathcal{A}$ can retrieve ground-truth labels from the gradient", how could $\mathcal{A}$ do that especially is there a state of the art methods that work with duplicate labels ?) 2. $\textbf{authors do not tackle the multiple updates case}$, which is characteristic of FL 3. the attack requires access to an external dataset 4. the attack can only work on classification tasks with numerous classes (more than the batch-size). Although 3 and 4 are standard assumptions in this literature, 1 and 2 are extremely contrived. This simplified setting is still of some interest but the claims do not match the reality. In addition, in the code repository, in the attack's code the signature of the function does not include the labels variable so executing the function would throw "labels is undefined" errors. This is a misrepresentation of the function in line with what is written in the article. In the same spirit when writing as defense increasing the batch-size it is more a limitation of the attack than a true defense deserving its own section. Furthermore there is no experiment in the main paper on the resolution of the images on the attack's performances whereas it is claimed the attack work on high-resolution images as well.

Throughout the paper, the authors write about the advantages of their method without properly highlighting its limitations. An example is "without imposing unrealistic assumptions about batch-size, number of classes, and label distribution": in fact the authors explain that their method can only work if the batch-size is lower than the number of classes (this is even the "defense" the authors propose) AND the attack can only work with full knowledge of the labels, which is completely unrealistic. So all such sentences should be replaced by more grounded and precise ones such as "Although our attack requires the full knowledge of labels, it can tackle any batch-sizes providing that the number of classes is higher than the batch-size. Our attack is also the first to be able to handle label duplication. " Another example is "Our approach reveals the vulnerability in cross-entropy loss function and fully-connected layer," whereas the author's approach is one of many many research work having explored this setting and the latest to date. The one article, which could have written this sentence is DLG (2019).

- The paper's writing is subpar. Most sentences are either too strong (see above section) or colloquial (use of the adjective "very" repeatedly (8 times), "$\textbf{we}$ establish the second equation" followed by a citation, overall sentences' constructions)  or not understandable at all (some of the typos / grammar mistakes are listed in Minor). Can the authors also define notations and acronyms when they are first introduced ?  For instance acronyms, DLG, FGLA, HCGLA, GGL, IG, etc. are not introduced. Neither are notations $y(i)$, $\nabla \theta^{W}\_{L, y(i)} $ $\mathcal{L}\_{1}$ and similar math notations. As $\mathcal{L}\_{1}$ is not defined "having lower $\mathcal{L}\_{1}$ values" does not mean anything and therefore he reviewer doesn't know how to read Figure 3. Most of the undefined notations are roughly understandable for readers familiar with the literature but it makes it very painful to read.

- The reviewer would like to see the performance of the attack (and authors' ideas about it) when handling realistic scenarii aka without at least knowing the ground truth labels. Is there a heuristic similar to GradInv [4] that can work ? Can some combinatorial approach be tried ? Also a plot in the main article showing the effect of increasing the number of updates on the performance of the attack is needed. Otherwise the authors should write that they attack FedSGD and not FL.


Minor
- Related works should be rewritten to have an entire paragraph (with a title) with attacks based on data priors (such as R-GAP that is mentioned)
- Use MacMahan's paper [2] for the FL reference instead of AhishekV et al. 2022 (authors can add Shokri's [3])
- Rewrite propositions 1 as $\hat{y}=...$
- Replace all "last" layer of FC by "first" layer of FC, usual convention is to count layers starting from the input. The feature map is thus the input of the first layer of the FC not the last. If the authors want to use a different convention the authors have to define it.
- typos: an analysis method page 3 ? / almost non-loss page 5 ? / will resolve page 4 / "When B < C, we cannot recover unique y with the established system of equations is overdetermined, thus there may be multiple solutions, of which only one is a right solution, and
the remaining solutions are referred to in prior work R-GAP (Zhu & Blaschko, 2020) as the twin solutions." this sentence is not correct page 6

[1] Kariyappa, Sanjay, et al. "Cocktail party attack: Breaking aggregation-based privacy in federated learning using independent component analysis." International Conference on Machine Learning. PMLR, 2023.

[2] McMahan, Brendan, et al. "Communication-efficient learning of deep networks from decentralized data." Artificial intelligence and statistics. PMLR, 2017.

[3] Shokri, Reza, and Vitaly Shmatikov. "Privacy-preserving deep learning." Proceedings of the 22nd ACM SIGSAC conference on computer and communications security. 2015.

[4] Yin, Hongxu, et al. "See through gradients: Image batch recovery via gradinversion." Proceedings of the IEEE/CVF Conference on Computer Vision and Pattern Recognition. 2021.

**Questions:**

The reviewer encourages the authors to
- rewrite the claims of the manuscript (specifically highlight the use of ground truth labels and remove the claims related to being the "first")
- improve overall grammar and notations. Especially explain what $\mathcal{L}\_{1}$ means so that the reviewer can read Figure 3.
- perform experiments 1. without ground truth labels 2. with high number of updates 3. varying resolutions to strenghten the contribution and close the gap between what is claimed and what is done
- explain more clearly if the other methods RGLA is compared to use ground truth labels as well ? If it is not the case the comparison is unfair if it is the case it should read DLG + ground truth labels etc. for most of those methods ,which do not require ground truth labels
- add a comparison to GradInv [4] removing the loss on BN statistics
- discuss the frequency of apparition of duplicate labels when randomly sampling batches from say IMAGENET

---

> ### Public Comment · ~Dimitar_Iliev_Dimitrov2 · 2023-11-10
> **Suggestion**
>
> I hope the reviewer doesn't mind me intervening. I think to relax the known label assumption, the authors can combine their work with one of the following:
> [1] Aidmar Wainakh, Fabrizio G. Ventola, Till Mußig, Jens Keim, Carlos Garcia Cordero, Ephraim Zimmer, Tim Grube, Kristian Kersting, and M. Muhlh ¨ auser. User-level label leakage from gradients in federated learning. Proceedings on Privacy Enhancing Technologies, 2022:227 – 244, 2021
> [2] Geng, Jiahui, et al. "Towards general deep leakage in federated learning." arXiv preprint arXiv:2110.09074 (2021).
> ( I have a somewhat working implementation in https://github.com/eth-sri/fedavg_leakage/blob/0edb1b948a9003c88b2f74d9105c68e22e90cc47/fedavg/utils/flax_losses.py#L339 but you can use your auxiliary data to fit it better.)
> [3] Kailang Ma, Yu Sun, Jian Cui, Dawei Li, Zhenyu Guan, and Jianwei Liu. 2023. Instance-wise batch label restoration via gradients in federated learning. In International Conference on Learning Representations

---

> ### Author Response · Authors · 2023-11-18
> **Thanks you for your suggestion**
>
> Thanks you for your suggestion, we combine the label inference technique of [3] with our method  to relax the label assumption and the experiment results can be seen in our rebuttal version.

---

> ### Author Response · Authors · 2023-11-18
> **Response to reviewer iafz (Part 1)**
>
> Thank you for your thorough review and valuable suggestions and questions on our manuscript. We sincerely accept your opinions and will make revisions and improvements. Note that the order of some figures/results has changed with the rebuttal revision. The implementation of the additional experiment can be found in our updated github repository and updated supplementary material. Below, we address specific points you raised.
>
> >**Q1 The lack of sufficient evidence supporting some claims, such as "the first to successfully disaggregate the feature map in a generic federated learning setting," and the omission of key reference [1] and one comparison with [4] removing the loss on BN statistics.**
>
> Thank you for bringing the works [1, 4] to our attention. We would like to emphasize that there are differences between them and our paper in terms of methods and results, and we have removed this statement in the revision. For [1], the Cocktail Party Attack (CPA) treats the problem of gradient inversion as a Blind Source Separation (BSS) issue, based on the premise that the gradients of a Fully Connected (FC) layer can be seen as a linear combination of its inputs. CPA then uses Independent Component Analysis (ICA) to address the BSS problem. In contrast, our method initially leaks the model's output and then further leaks the feature map. This strategy allows all methods that invert model inputs based on model outputs to be applicable in the context of gradient leakage attacks, thereby significantly increasing the vulnerability of gradients. We discuss this paper in the related works section of our rebuttal version.
>
> Since there is no official source code for the previous work [4] (STG), we perform the STG using the STG source code published by FGLA [8], which we also included in our open-source code. We compared our method with STG (with or without BN statistics) using the ImageNet dataset (224$\times$224) and batch size of 8, and the results are shown in Table 1. It is clear from Table 1 that our method outperforms prior work [4] in terms of time efficiency and reconstruction results.
>
> | Method                    | PSNR$\uparrow$ | SSIM$\uparrow$ | LPIPS$\downarrow$ | Time$\downarrow$(s) |
> | ------------------------- | -------------- | -------------- | ----------------- | ------------------- |
> | Our RGLA                  | 19.24153       | 0.50947        | 0.55870           | 14.35               |
> | STG with BN statistics    | 11.47697       | 0.35237        | 0.66932           | 9484.70             |
> | STG without BN statistics | 9.90506        | 0.32183        | 0.89051           | 7497.48             |
>
> Table 1
>
> >**Q2 1. Assumption of ground-truth labels: the attack needs access to ground truth labels, which differs from many competing methods, and queries how $\mathcal{A}$ could feasibly retrieve these labels from gradients. 2. Omission of multiple updates scenario in FL: the proposed method doesn't address the characteristic feature of multiple updates in FL. 3. Attack's dependence on an external dataset. 4. Limitation to classification tasks of B<=C. In the same spirit when writing as defense increasing the batch-size it is more a limitation of the attack than a true defense deserving its own section. 5. Code repository has a bug. 6. Lack of experiments on varying resolution images.**
>
> 1. We would like to clarify that the assumption of ground-truth labels is reasonable for the reason. This label assumption shared in recent studies [7-10], which either assume that the adversary knows the labels [7, 8], or first extract the labels [9, 10, 11] using existing label inference techniques.
>
>    To improve rigor, we combine our method with a stat-of-the-art label inference technique, iLRG [6]. Table 2 shows the average results of data reconstruction on 100 batches. Notably, iLRG achieved a 100% accuracy rate in our experiments, though the order of the reconstructed labels differs from that of the actual ground-truth labels. As shown in Table 2, the reconstructed results obtained using ground-truth labels and those reconstructed results using the iLRG [6] are similar. Fig. 11 in our rebuttal version provides visual reconstruction results, illustrating that both ground-truth labels and inferred labels can successfully reconstruct visual outcomes, except that using inferred labels causes the reconstructed data to be in a different order from the original data. We also add this experiment and corresponding discussion in the Appendix of our rebuttal version.
>
>    |                                 | PSNR$\uparrow$ | SSIM$\uparrow$ | LPIPS$\downarrow$ |
>    | ------------------------------- | -------------- | -------------- | ----------------- |
>    | Ground-truth label + RGLA(Ours) | 19.24153       | 0.50947        | 0.55870           |
>    | iLRG[6]+ RGLA(Ours)             | 19.22677       | 0.49973        | 0.56795           |
>
>    Table 2

---

> > ### Author Response · Authors · 2023-11-18
> > **Response to reviewer iafz (Part 2)**
> >
> > 2. Like most current works [8, 11, 12, 13, 14, 15, 16, 17, 18, 19, 20] in our research field, we also focus on gradient leakage attacks in FedSGD. Consequently, we did not specifically emphasize FedSGD in our original manuscript like these works. However, we agree with the importance of your observation in enhancing the rigor of our manuscript. Therefore, in the rebuttal version, we have explicitly emphasized FedSGD in our threat model, clarifying that our manuscript focuses on the gradient of one update.
> >
> > 3. Accessing an additional dataset. Many of the more recent works [7, 8, 11, 12, 20, 21] in our research field require access to an external dataset. Moreover, we believe that in a generic federated learning scenario, an attacker can use some open-source datasets as auxiliary data to facilitate the attack.
> >
> > 4. We believe that in a generic federated learning scenario, there are instances where the batch size is smaller than the number of classification classes. Additionally, some uninformed victim clients may not be aware of the need to increase the batch size more than the number of classes. Furthermore, for classification tasks with a particularly large number of categories, it may not be feasible to increase the batch size beyond the number of classes due to hardware limitations. With your suggestion in mind, we have written the B<=C as a limitation of our method in the conclusion section of the rebuttal version.
> >
> > 5. We have tested our code on different machines, but do not encounter your mentioned error. Could you provide detailed error information? We will do our best to help you solve it.
> >
> > 6. In our experiments, to maintain consistency and avoid the need for training multiple generators, we scaled all images to a resolution of 224$\times$224 pixels. This resolution choice aligns with the prior works in our field, such as [1, 4], which also regard 224$\times$224 pixels as high resolution. We haven't seen any prior works that can work with larger pixels than 224$\times$224. But, we find your comment valuable and we conducted experiments on images of 336$\times$336 pixels, for which we adapt the structure of the generator to accommodate different sizes of inputs and outputs and train it. The experiment results in Table 3 show that our method is able to reconstruct the original private data regardless of whether the original data is 224$\times$224 pixels or 336$\times$336 pixels. The decrease in the quality of the reconstruction results for the original data of 336$\times$336 pixels can be attributed to the fact that the 336$\times$336 pixels image is more complex and poses a greater challenge to the reconstruction capability of the generator. As a result, the intricate details of such complex images cannot be accurately captured during the reconstruction process, leading to overall quality degradation. Fig. 12 in our rebuttal version shows the visual reconstruction results of the same batch of images of different resolutions. These visualizations clearly show that our method maintains its effectiveness in reconstructing the image batch visually despite the high resolution, demonstrating the robustness of our method at high pixels. We have added this experiment and related discussion to our rebuttal version.
> >
> >    |                      | PSNR$\uparrow$ | SSIM$\uparrow$ | LPIPS$\downarrow$ | Time$\downarrow$ |
> >    | -------------------- | -------------- | -------------- | ----------------- | ---------------- |
> >    | RGLA(224$\times$224) | 18.80249       | 0.496655       | 0.588019          | 14.893825        |
> >    | RGLA(336$\times$336) | 16.75473       | 0.466147       | 0.380592          | 14.988002        |
> >
> >       Table 3

---

> > > ### Author Response · Authors · 2023-11-18
> > > **Response to reviewer iafz (Part 3)**
> > >
> > > >**Q3 Throughout the paper, the authors write about the advantages of their method without properly highlighting its limitations. An example is "without imposing unrealistic assumptions about batch size, number of classes, and label distribution": in fact, the authors explain that their method can only work if the batch size is lower than the number of classes (this is even the "defense" the authors propose) and the attack can only work with full knowledge of the labels, which is completely unrealistic. So all such sentences should be replaced by more grounded and precise ones such as "Although our attack requires the full knowledge of labels, it can tackle any batch size providing that the number of classes is higher than the batch size. Our attack is also the first to be able to handle label duplication. " Another example is "Our approach reveals the vulnerability in cross-entropy loss function and fully-connected layer," whereas the author's approach is one of many research works that have explored this setting and the latest to date. The one article, which could have written this sentence is DLG (2019).**
> > >
> > > Thank you very much for your suggestion! Our mention in the manuscript of "without imposing unrealistic assumptions about batch size, number of classes, and label distribution" stems from our belief that such scenarios that B<=C+1 do exist in practical federated learning settings, particularly in tasks with a large number of classification classes. We consider this just an assumption, not an unrealistic assumption.  However, we acknowledge your suggestion for more precise articulation and have accordingly revised our statement in our rebuttal version. We improved the sentence "Our approach reveals the vulnerability in cross-entropy loss function and fully-connected layer" in our rebuttal version.
> > >
> > > >**Q4 1. Most sentences are either too strong (see above section) or colloquial (use of the adjective "very" repeatedly (8 times), "we establish the second equation" followed by a citation, overall sentences' constructions), or not understandable at all (some of the typos/grammar mistakes are listed in Minor). 2. Can the authors also define notations and acronyms when they are first introduced? For instance, acronyms, DLG, FGLA, HCGLA, GGL, IG, etc. are not introduced. Neither are notations $y(i)$, $\nabla {\theta_ {L}^{b}}_ {y_ {(i)}}$,$\mathcal{L}_ 1$ and similar math notations. As $\mathcal{L}_ 1$ is not defined "having lower $\mathcal{L}_ 1$ values" does not mean anything and therefore the reviewer doesn't know how to read Figure 3.**
> > >
> > > Thank you very much for your suggestion!
> > >
> > > 1. We improved our expressions in the rebuttal version.
> > >
> > > 2. In the rebuttal version, we introduce acronyms such as DLG (Deep leakage attack), etc. $y(i)$ denotes the label of the $i^{th}$ sample, and $\nabla {\theta_ {L}^{b}}_ {y_ {(i)}}$ denotes the $y(i)^{th}$ row of the gradient of the bias layer, and the definition of $\mathcal{L}_ 1$ is in Equation 4 on the fifth page of the main text of the manuscript.
> > >
> > > >**Q5 1. Use MacMahan's paper [2] for the FL reference instead of AhishekV et al. 2022 (authors can add Shokri's [3]). 2. Replace all "last" layers of FC with the "first" layer of FC, the usual convention is to count layers starting from the input. The feature map is thus the input of the first layer of the FC not the last. If the authors want to use a different convention the authors have to define it. 3. Typos: an analysis method page 3 ? / almost non-loss page 5? / will resolve page 4 / "When B < C, we cannot recover unique y with the established system of equations is overdetermined, thus there may be multiple solutions, of which only one is a right solution, and the remaining solutions are referred to in prior work R-GAP (Zhu & Blaschko, 2020) as the twin solutions." this sentence is not correct page 6.**
> > >
> > > 1. We use MacMahan's paper [2] for the FL reference instead of AhishekV et al. 2022 and add Shokri's [3] in the rebuttal version.
> > > 2. In our rebuttal version, we have defined our feature map to be the input to the last fully connected layer
> > > 3. The typos mentioned have been corrected in the latest version, which you can download and check out!
> > >
> > > >**Q6  Explain more clearly if the other methods compared to RGLA use ground truth labels as well. If it is not the case the comparison is unfair if it is the case it should read DLG + ground truth labels etc. for most of those methods, which do not require ground truth labels.**
> > >
> > > Thank you for your suggestion! We compared DLG, IG, GGL, and FGLA in our comparison experiments, where DLG, IG, and FGLA used ground-truth labels, and GGL did not use ground-truth labels but inferred labels by the iDLG method. We reperform our experiments using GGL + ground-truth labels and updated our experimental results in the rebuttal version. We highlight the use of ground-truth labels for these methods in the comparison subsection in the rebuttal version.

---

> > > > ### Author Response · Authors · 2023-11-18
> > > > **Response to reviewer iafz (Part 4)**
> > > >
> > > > >**Q7 Discuss the frequency of apparition of duplicate labels when randomly sampling batches from say IMAGENET.**
> > > >
> > > > The ImageNet dataset has a total of 1000 classes, and assuming a batch size of $B$, the probability of existing of a duplicate label is $1-(\frac{1000}{1000}\times\frac{999}{1000}\times\frac{998}{1000}...\times\frac{1000-B+1}{1000})$.
> > > >
> > > > Reference:
> > > >
> > > > [1] Kariyappa, Sanjay, et al. "Cocktail party attack: Breaking aggregation-based privacy in federated learning using independent component analysis." International Conference on Machine Learning. PMLR, 2023.
> > > >
> > > > [2] McMahan, Brendan, et al. "Communication-efficient learning of deep networks from decentralized data." Artificial intelligence and statistics. PMLR, 2017.
> > > >
> > > > [3] Shokri, Reza, and Vitaly Shmatikov. "Privacy-preserving deep learning." Proceedings of the 22nd ACM SIGSAC conference on computer and communications security. 2015.
> > > >
> > > > [4] Yin, Hongxu, et al. "See through gradients: Image batch recovery via gradinversion." Proceedings of the IEEE/CVF Conference on Computer Vision and Pattern Recognition. 2021.
> > > >
> > > > [5] Wainakh A, Ventola F, Müßig T, et al. User-level label leakage from gradients in federated learning[J]. arXiv preprint arXiv:2105.09369, 2021.
> > > >
> > > > [6] Ma K, Sun Y, Cui J, et al. Instance-wise Batch Label Restoration via Gradients in Federated Learning[C]//The Eleventh International Conference on Learning Representations. 2022.
> > > >
> > > > [7] Jinwoo Jeon, Jaechang Kim, Kangwook Lee, Sewoong Oh, and Jungseul Ok. Gradient inversion with generative image prior. In Neural Information Processing Systems, pp. 29898–29908, 2021.
> > > >
> > > > [8] Dongyun Xue, Haomiao Yang, Mengyu Ge, Jingwei Li, Guowen Xu, and Hongwei Li. Fast generation-based gradient leakage attacks against highly compressed gradients. IEEE INFOCOM 2023 - IEEE Conference on Computer Communications, 2023.
> > > > [9] Zhu J, Blaschko M. R-gap: Recursive gradient attack on privacy[J]. arXiv preprint arXiv:2010.07733, 2020.
> > > >
> > > > [10] Jonas Geiping, Hartmut Bauermeister, Hannah Droge, and Michael Moeller. Inverting gradients - ¨how easy is it to break privacy in federated learning? ArXiv, abs/2003.14053, 2020.
> > > >
> > > > [11] Yang H, Ge M, Xiang K, et al. Using Highly Compressed Gradients in Federated Learning for Data Reconstruction Attacks[J]. IEEE Transactions on Information Forensics and Security, 2022, 18: 818-830.
> > > >
> > > > [12] Zhuohang Li, Jiaxin Zhang, Luyang Liu, and Jian Liu. Auditing privacy defenses in federated learning via generative gradient leakage. In Proceedings of the IEEE/CVF Conference on Computer Vision and Pattern Recognition, pp. 10132–10142, 2022.
> > > >
> > > > [13] Zhao B, Mopuri K R, Bilen H. idlg: Improved deep leakage from gradients[J]. arXiv preprint arXiv:2001.02610, 2020.
> > > >
> > > > [14] Ligeng Zhu, Zhijian Liu, and Song Han. Deep leakage from gradients. In Neural Information Processing Systems, pages 17–31, 2019.
> > > >
> > > > [15] Le Trieu Phong, Yoshinori Aono, Takuya Hayashi, Lihua Wang, and Shiho Moriai. Privacy preserving deep learning: Revisited and enhanced. In International Conference on Applications and Techniques in Information Security, 2017.
> > > >
> > > > [16] Lixin Fan, Kam Woh Ng, Ce Ju, Tianyu Zhang, Chang Liu, Chee Seng Chan, and Qiang Yang. Rethinking privacy preserving deep learning: How to evaluate and thwart privacy attacks. ArXiv, abs/2006.11601, 2020.
> > > >
> > > > [17] Liam Fowl, Jonas Geiping, Wojtek Czaja, Micah Goldblum, and Tom Goldstein. Robbing the fed: Directly obtaining private data in federated learning with modified models. arXiv preprint arXiv:2110.13057, 2021.
> > > >
> > > > [18] Franziska Boenisch, Adam Dziedzic, Roei Schuster, Ali Shahin Shamsabadi, Ilia Shumailov, and Nicolas Papernot. When the curious abandon honesty: Federated learning is not private. arXiv preprint arXiv:2112.02918, 2021.
> > > >
> > > > [19] Briland Hitaj, Giuseppe Ateniese, and Fernando Perez-Cruz. Deep models under the gan: information leakage from collaborative deep learning. In Proceedings of the 2017 ACM SIGSAC 357 conference on computer and communications security, pages 603–618, 2017.
> > > >
> > > > [20] Hanchi Ren, Jingjing Deng, and Xianghua Xie. Grnn: generative regression neural network—a data leakage attack for federated learning. ACM Transactions on Intelligent Systems and Technology (TIST), 13(4):1–24, 2022.
> > > >
> > > > [21] Yue K, Jin R, Wong C W, et al. Gradient obfuscation gives a false sense of security in federated learning[C]//32nd USENIX Security Symposium (USENIX Security 23). 2023: 6381-6398.

---

> > > > > ### Comment · Reviewer_iafz · 2023-11-20
> > > > >
> > > > > Computing the probability from which the authors find the formula for B=256 or the target batch-size would lead to a either low or high frequency of duplicate labels. This was the reasoning behind the comment of the reviewer. Knowing whether or not one is expected to encounter many duplicate labels is important.

---

> > > ### Comment · Reviewer_iafz · 2023-11-20
> > >
> > > Regarding 5. to illustrate more explicitly the problem, the current code in the README reads:
> > > def Model_output_leakage(grad, model):
> > > This gives the false impression that it does not need labels, that are accessed here wo being defined in the function:
> > > pred_modelloss = pred_loss(grad, label, batchsize, defence_method)
> > > Label is declared globally which is a bad code practice in addition the correct signature should read:
> > > def Model_output_leakage(grad, model, labels):
> > > This example is representative of the rest of the paper.

---

> ### Public Comment · ~Dimitar_Iliev_Dimitrov2 · 2023-11-19
> **Re: Q2**
>
> **Response to Q2**: Can the authors comment on their attack's compatibility with FedAvg? Can they experiment with attacking the FedAvg setting? In particular, here [1] will be useful, as it might be easy to combinable with the author's method.
> [1] Zhu, Junyi, Ruicong Yao, and Matthew B. Blaschko. "Surrogate model extension (SME): A fast and accurate weight update attack on federated learning." arXiv preprint arXiv:2306.00127 (2023).

---

> > ### Author Response · Authors · 2023-11-22
> > **Regarding the combination of RGLA and SME to be applicable in a FedAvg**
> >
> > Thank you for bringing [1] to our attention. We believe that our method can be combined with [1] to be applicable in a FedAvg scenario. We plan to explore this combination after the rebuttal period, as there are too many derivations to combine for the combination of these two works. Should the results prove promising, we intend to include this combination in the Appendix in the latest version of our paper. Thanks for your attention to our manuscript.
> >
> > [1] Zhu, Junyi, Ruicong Yao, and Matthew B. Blaschko. "Surrogate model extension (SME): A fast and accurate weight update attack on federated learning." arXiv preprint arXiv:2306.00127 (2023).

---

> > > ### Public Comment · ~Dimitar_Iliev_Dimitrov2 · 2023-11-22
> > > **Regarding the combination of RGLA and SME to be applicable in a FedAvg**
> > >
> > > Thank you for considering this. I think, given the prevalence of FedAvg, it can further boost the claims of the paper.

---

> ### Comment · Reviewer_iafz · 2023-11-20
> **Final comment**
>
> While the reviewer appreciate the dedication of the authors during the rebuttal process and the thorough answers. The reviewer stand with their initial scoring.
>
> The authors are misrepresenting their findings (I take the current answer to question 5 as one of the many examples of misrepresenting results: "We have tested our code on different machines, but do not encounter your mentioned error. Could you provide detailed error information? We will do our best to help you solve it." while the fact that the function's code on the README is blatantly misleading is difficult to refute) and thus it makes it hard to form an accurate picture of the contribution as the findings of the rebuttal would need to be reviewed with more care. Notably the addition of experiments on GradInv brings more questions than it answers.
> The authors also disregarded in the paper the two most related works GradInv and BSS in the literature, which, considering the rest of the paper, is interpreted by the reviewer as not completely genuine.
>
> A minima, the reviewer would like to see the authors' paper rewritten more clearly by either 1. potentially redoing all experiments with iLRG and dropping the known labels assumption 2. stating the assumption in a less confusing manner in the text and in the code not to misrepresent the contribution. Cleaning the text from the abundance of overstatements and misleading statements.
>
> The reviewer also disagrees with the assessment from reviewer p8x8.

---

> ### Author Response · Authors · 2023-11-21
> **Improvement on the function declarations in the README file in our code**
>
> >**Q8 Regarding 5. to illustrate more explicitly the problem, the current code in the README reads: def Model_output_leakage(grad, model): This gives the false impression that it does not need labels, that are accessed here wo being defined in the function: pred_modelloss = pred_loss(grad, label, batchsize, defence_method) Label is declared globally which is a bad code practice in addition the correct signature should read: def Model_output_leakage(grad, model, labels): This example is representative of the rest of the paper.**
>
> Thank you for pointing this out, we have improved the function declarations in the README file in our code. The introduction, threat model, method, and experiment section of our manuscript explicitly state that our approach requires the ground-truth label, an assumption also shared with many well-known works [1] [2] [3] [4] [5] [6] [7] [8] [9] [10] in this research field.
>
>
> [1] Zhu J, Blaschko M. R-gap: Recursive gradient attack on privacy[J]. arXiv preprint arXiv:2010.07733, 2020.
>
> [2] Jonas Geiping, Hartmut Bauermeister, Hannah Droge, and Michael Moeller. Inverting gradients - ¨how easy is it to break privacy in federated learning? ArXiv, abs/2003.14053, 2020.
>
> [3] Hongxu Yin, Pavlo Molchanov, Jose M. Alvarez, Zhizhong Li, Arun Mallya, Derek Hoiem, Niraj K Jha, and Jan Kautz. Dreaming to distill: Data-free knowledge transfer via DeepInversion. In CVPR, 2020. 2, 5, 6.
>
> [4] Yin, Hongxu, et al. "See through gradients: Image batch recovery via gradinversion." Proceedings of the IEEE/CVF Conference on Computer Vision and Pattern Recognition. 2021.
>
> [5] Jinwoo Jeon, Jaechang Kim, Kangwook Lee, Sewoong Oh, and Jungseul Ok. Gradient inversion with generative image prior. In Neural Information Processing Systems, pp. 29898–29908, 2021.
>
> [6] Yang H, Ge M, Xiang K, et al. Using Highly Compressed Gradients in Federated Learning for Data Reconstruction Attacks[J]. IEEE Transactions on Information Forensics and Security, 2022, 18: 818-830.
>
> [7] Kariyappa, Sanjay, et al. "Cocktail party attack: Breaking aggregation-based privacy in federated learning using independent component analysis." International Conference on Machine Learning. PMLR, 2023.
>
> [8] Dongyun Xue, Haomiao Yang, Mengyu Ge, Jingwei Li, Guowen Xu, and Hongwei Li. Fast generation-based gradient leakage attacks against highly compressed gradients. IEEE INFOCOM 2023 - IEEE Conference on Computer Communications, 2023.
>
> [9] Yue K, Jin R, Wong C W, et al. Gradient obfuscation gives a false sense of security in federated learning[C]//32nd USENIX Security Symposium (USENIX Security 23). 2023: 6381-6398.
>
> [10] Zhu J, Yao R, Blaschko M B. Surrogate model extension (SME): A fast and accurate weight update attack on federated learning[J]. arXiv preprint arXiv:2306.00127, 2023.

---

> ### Author Response · Authors · 2023-11-21
> **Regarding the probability of occurrence of duplicate labels**
>
> > **Q9 Computing the probability from which the authors find the formula for B=256 or the target batch size would lead to a either low or high frequency of duplicate labels. This was the reasoning behind the comment of the reviewer. Knowing whether or not one is expected to encounter many duplicate labels is important.**
>
> The ImageNet dataset has a total of 1000 classes, and assuming a batch size of $B$, the probability of the existence of a duplicate label is:
>
> $p=1-(\frac{1000}{1000}\times\frac{999}{1000}\times\frac{998}{1000}...\times\frac{1000-B+1}{1000})$.
>
> Therefore, when the batch size is set to 256, the probability of encountering duplicate labels is 0.99999. We have also created a graph illustrating how the probability \( $p$ \) varies with the batch size \( $B$ \). You can find this graph in the supplementary materials under the filename "DuplicateLabelProbability.png".
>
> Also, if you are interested in seeing the performance of our proposed method under different proportions of duplicate labels within a batch, please refer to Section 5.1 of our manuscript.

---

> ### Author Response · Authors · 2023-11-21
> **Response to your final comment (Part 1)**
>
> >**Q10 The authors are misrepresenting their findings (I take the current answer to question 5 as one of the many examples of misrepresenting results: "We have tested our code on different machines, but do not encounter your mentioned error. Could you provide detailed error information? We will do our best to help you solve it." while the fact that the function's code on the README is blatantly misleading is difficult to refute) and thus it makes it hard to form an accurate picture of the contribution as the findings of the rebuttal would need to be reviewed with more care.**
>
> Sorry, we thought your concern was about a bug in our code. Thank you for pointing this out, and in order to make our contribution clearer, we've improved the README file in our code, which hopefully addresses your concern!
>
> >**Q11 Notably the addition of experiments on GradInv brings more questions than it answers.**
>
> We are seeking clarification regarding your statement, "Notably the addition of experiments on GradInv [4] brings more questions than it answers." Are you suggesting that there is an issue with our implementation of GradInv [4]? Considering that GradInv [4] does not have an official open-source code, we adopted the implementation of GradInv [4] as used in FGLA, who compared it in their work. We believe their implementation can be considered reliable. Alternatively, you may run the code we have provided in the supplementary materials for further verification.
>
> >**Q12 The authors also disregarded in the paper the two most related works GradInv and BSS in the literature, which, considering the rest of the paper, is interpreted by the reviewer as not completely genuine.**
>
> We have discussed both GradInv [4] and BSS [7] in the related work section of our manuscript and have accordingly adjusted the rest of the paper.
>
> >**Q13 A minima, the reviewer would like to see the authors' paper rewritten more clearly by either 1. potentially redoing all experiments with iLRG and dropping the known labels assumption 2. stating the assumption in a less confusing manner in the text and in the code not to misrepresent the contribution. Cleaning the text from the abundance of overstatements and misleading statements.**
>
> Thank you for your time and effort in providing such detailed and important feedback. We believe our assumption about labels is reasonable, and below, we explain why this assumption is reasonable.
>
> 1. Firstly, existing techniques for inferring labels are quite mature and can reconstruct true labels with high accuracy, up to 100%. Therefore, we assume adversaries can obtain true labels, allowing us to focus our limited paper space on the theft of private data, which is likely the primary concern for adversaries.
>
> 2. Second, we have had in-depth discussions on this issue with researchers in the field. Indeed, most of the notable work [1] [2] [3] [4] [5] [6] [7] [8] [9] [10] in our research area is based on a common assumption - that labels are accessible. This general consensus supports our approach and allows us to focus our main efforts on the core problem of data theft.
>
> 3. Lastly, our experiments on the ImageNet dataset and the ResNet50 model show that when the batch size is less than 256, iLRG[11] achieves 100% accuracy in label reconstruction (also as evident from Figure 3 in [11]). This indicates that adversaries can easily obtain true labels. Furthermore, all the state-of-the-art methods compared in our paper also used ground-truth labels, making our experimental comparisons fair.
>
> Considering these points, the assumption that adversaries can access ground-truth labels is reasonable.
>
> We have adjusted our manuscript to state assumptions and contributions more clearly. Please refer to the latest rebuttal version for these changes.

---

> > ### Author Response · Authors · 2023-11-21
> > **Response to your final comment (Part 2)**
> >
> > Reference:
> >
> > [1] Zhu J, Blaschko M. R-gap: Recursive gradient attack on privacy[J]. arXiv preprint arXiv:2010.07733, 2020.
> >
> > [2] Jonas Geiping, Hartmut Bauermeister, Hannah Droge, and Michael Moeller. Inverting gradients - ¨how easy is it to break privacy in federated learning? ArXiv, abs/2003.14053, 2020.
> >
> > [3] Hongxu Yin, Pavlo Molchanov, Jose M. Alvarez, Zhizhong Li, Arun Mallya, Derek Hoiem, Niraj K Jha, and Jan Kautz. Dreaming to distill: Data-free knowledge transfer via DeepInversion. In CVPR, 2020. 2, 5, 6.
> >
> > [4] Yin, Hongxu, et al. "See through gradients: Image batch recovery via gradinversion." Proceedings of the IEEE/CVF Conference on Computer Vision and Pattern Recognition. 2021.
> >
> > [5] Jinwoo Jeon, Jaechang Kim, Kangwook Lee, Sewoong Oh, and Jungseul Ok. Gradient inversion with generative image prior. In Neural Information Processing Systems, pp. 29898–29908, 2021.
> >
> > [6] Yang H, Ge M, Xiang K, et al. Using Highly Compressed Gradients in Federated Learning for Data Reconstruction Attacks[J]. IEEE Transactions on Information Forensics and Security, 2022, 18: 818-830.
> >
> > [7] Kariyappa, Sanjay, et al. "Cocktail party attack: Breaking aggregation-based privacy in federated learning using independent component analysis." International Conference on Machine Learning. PMLR, 2023.
> >
> > [8] Dongyun Xue, Haomiao Yang, Mengyu Ge, Jingwei Li, Guowen Xu, and Hongwei Li. Fast generation-based gradient leakage attacks against highly compressed gradients. IEEE INFOCOM 2023 - IEEE Conference on Computer Communications, 2023.
> >
> > [9] Yue K, Jin R, Wong C W, et al. Gradient obfuscation gives a false sense of security in federated learning[C]//32nd USENIX Security Symposium (USENIX Security 23). 2023: 6381-6398.
> >
> > [10] Zhu J, Yao R, Blaschko M B. Surrogate model extension (SME): A fast and accurate weight update attack on federated learning[J]. arXiv preprint arXiv:2306.00127, 2023.
> >
> > [11] Ma K, Sun Y, Cui J, et al. Instance-wise Batch Label Restoration via Gradients in Federated Learning[C]//The Eleventh International Conference on Learning Representations. 2022.

---

### Official Review · Reviewer_nX92 · 2023-10-30

**Soundness:** 3 good
**Presentation:** 2 fair
**Contribution:** 2 fair
**Rating:** 6
**Confidence:** 3

**Summary:**

This paper studies gradient leakage attacks in federated learning. Motivated by the inefficacy of existing attacks against large batches of high-resolution images, the authors propose a new attack named reverse gradient leakage attack (RGLA). RGLA involves three stages: first inverts the cross-entropy loss function to obtain the model outputs, which are then disaggregated and inverted to feature maps. Finally, these feature maps are inverted to model inputs, leveraging a pre-trained generator model. Experiments on four datasets (ImageNet, Cifar-10, Cifar-100, and CelebA) verified the effectiveness of the proposed attack.

**Strengths:**

- The proposed attack can be applied to recover large batches of high-resolution images (e.g., 224x224px) with potentially duplicated labels.

- The proposed attack has a much smaller search space compared to optimization-based methods.

- The proposed attack remains effective against highly compressed gradients with added noise.

- Evaluations and comparisons with other attacks validate the value of the proposed attack. The reviewer was particularly impressed by the visualized reconstruction results for a batch of 256 images.

**Weaknesses:**

- The proposed method is built on several existing techniques that made disparate adversarial assumptions. As a result, the proposed RGLA combining these techniques requires a quite restrictive threat model, e.g., access to auxiliary datasets and ground-truth labels. In particular, RGLA requires the target network to have no pooling layer, which is hard to justify in practice. On the other hand, RGLA does not need batch statistics for reconstructing batched input but does seem to require B to not exceed C+1.

- The technical contributions of this work are not particularly clear. The core techniques adopted by RGLA for enabling large batch recovery (e.g., disaggregating gradients and training a generator model) were discovered in prior work by Xue et al. and the inversion on the model output was discussed by Zhu & Blaschko et al. The only distinction seems to be the relaxation on duplicate labels. Besides, some closely related works were not compared/discussed. For instance, [1] also trains a model to learn inverse mapping, and the idea of disaggregating and inverting feature vectors is very similar to the cocktail party attack [2].

- The proof of Proposition 2 made assumptions about the expressivity of the model, which should be made explicit in the text.

- The image similarity-based privacy metrics have some inherent limitations. For instance, the reconstructed images on CelebA have high measured PSNR but barely reveal any practically identifiable information. It would be better to add corresponding discussions.

[1] Wu, Ruihan, et al. "Learning to invert: Simple adaptive attacks for gradient inversion in federated learning." Uncertainty in Artificial Intelligence. PMLR, 2023.

[2] Kariyappa, Sanjay, et al. "Cocktail party attack: Breaking aggregation-based privacy in federated learning using independent component analysis." International Conference on Machine Learning. PMLR, 2023.

**Questions:**

1. From what the reviewer understands, the performance degradation of exiting attacks in the duplicated label scenario is an artifact of the label ambiguity. If that’s the case, existing optimization-based attacks should perform as well as if there were no duplicated labels in the extreme case where all images come from the same class, but that’s not what’s observed in Table 8 - existing attacks still perform poorly even if there is no label ambiguity. What causes the performance of existing attacks to drop? How would these methods perform if the true labels are assumed to be known?

2. What learning task is considered for experiments on the CelebA dataset? How many classes are there? What is the auxiliary dataset used?

3. The elimination of the twin solution seems to rely on its smaller loss value (as empirically verified in Fig. 3). How are these twin data eliminated on a well-trained model where all training data have relatively small loss values?

4. As RGLA relies on inverting the model output to feature maps, how would defense methods that perturb/modify the FCL part of the model (e.g., [1][2]) affect the attack performance?

5. It is interesting to see that gradient clipping and additive noise have little effect on FGLA as combining these two essentially provides some notion of differential privacy. What is the largest epsilon value that is able to defend FGLA and the proposed RGLA attack?

[1] Sun, Jingwei, et al. "Soteria: Provable defense against privacy leakage in federated learning from representation perspective." Proceedings of the IEEE/CVF conference on computer vision and pattern recognition. 2021.

[2]Scheliga, Daniel, Patrick Mäder, and Marco Seeland. "Precode-a generic model extension to prevent deep gradient leakage." Proceedings of the IEEE/CVF Winter Conference on Applications of Computer Vision. 2022.

---

> ### Author Response · Authors · 2023-11-18
> **Response to reviewer nX92 (Part 1)**
>
> Thank you for your thorough review and valuable suggestion and question on our manuscript. We sincerely accept your opinions and will make revisions and improvements. Note that the order of some figures/results has changed with the rebuttal revision. The implement of the additional experiment can be find in our updated github repository and updated supplementary material. Below, we address specific points you raised.
>
> >**Q1 1. Assumption of auxiliary datasets 2. Assumption of ground-truth labels. 3. Requirement on the target network to have no pooling layer. 4. Requirement on B to not exceed C+1.**
>
> 1. Accessing an auxiliary dataset: many of the more recent works [1, 2, 3, 4, 5, 6] in our research field require access to an external dataset. Moreover, we believe that in a real federated learning scenario, an adversary can use some open-source datasets as auxiliary data to facilitate the attack, which is not a difficult thing.
>
> 2. Access to ground-truth labels. We would like to clarify that the assumption of ground-truth labels is reasonable for this reason. This label assumption shared in recent studies [1, 2, 4, 9, 10], which either assume that the adversary knows the labels [1, 2], or first extract the labels [4, 9, 10] using existing label inference techniques.
>
>    To improve rigor, we combine our method with a state-of-the-art label inference technique, iLRG [8]. Table 1 shows the average results of data reconstruction on 100 batches. Notably, iLRG achieved a 100% accuracy rate in our experiments, though the order of the reconstructed labels differs from that of the actual ground-truth labels. As shown in Table 1, the reconstructed results obtained using ground-truth labels and those reconstructed results using the iLRG [8] are similar. Fig. 11 in our rebuttal version provides visual reconstruction results, illustrating that both ground-truth labels and inferred labels can successfully reconstruct visual outcomes, except that using inferred labels causes the reconstructed data to be in a different order from the original data. We also add this experiment and corresponding discussion in the Appendix of our rebuttal version.
>
>    |                                 | PSNR$\uparrow$ | SSIM$\uparrow$ | LPIPS$\downarrow$ |
>    | ------------------------------- | -------------- | -------------- | ----------------- |
>    | Ground-truth label + RGLA(Ours) | 19.24153       | 0.50947        | 0.55870           |
>    | iLRG [8] + RGLA(Ours)           | 19.22677       | 0.49973        | 0.56795           |
>
>    Table 1
>
> 3. The target network to have no pooling layer: Indeed, we removed the average pooling layer over the fully-connected layer (FC layer) in our experiments in order to make the feature map larger so that it contains more information about the original data, which is consistent with what was done in [2]. In order to explore how much information is lost by the average pooling layer, we conducted experiments in two scenarios: one in which the pooling layer is removed and the other in which the pooling layer is retained. And keep the rest of the experimental settings the same, including a batch size of 8, a dataset of ImageNet, and attacking the same 100 data batches. According to the experimental results in Table 2, whether the pooling layer is removed or not, our proposed method is able to reconstruct model outputs that are very close to the actual model outputs, which further reconstructs feature maps that are similar to the original feature maps. However, in the setting with the pool layer, the generator can not generate the original data even with the precise feature map. This phenomenon can be attributed to the fact that the addition of the pooling layer reduces the information about the original data contained in the feature maps, making it difficult for the generator to reconstruct the original image from feature maps that contain less information about the original data. The contribution of our manuscript does not lie in the training of the generator, which is the main contribution of [1]. In the future, we also expect more powerful generator models to be able to reconstruct the original images from corresponding feature maps even with limited information contained.

---

> > ### Author Response · Authors · 2023-11-18
> > **Response to reviewer nX92 (Part 2)**
> >
> > |                               | $\left\|\hat{y}^{\prime} - \hat{y} \right\|^2$ $\downarrow$ | $\left\|f^{\prime} - f \right\|^2$ $\downarrow$ | PSNR $\uparrow$ | SSIM $\uparrow$ | LPIPS $\downarrow$ |
> >    | ----------------------------- | ----------------------------------------------------------- | ----------------------------------------------- | --------------- | --------------- | ------------------ |
> >    | Without average pooling layer | 2.00E-04                                                    | 1.01E-04                                        | 18.72215        | 0.48819         | 0.57847            |
> >    | With average pooling layer    | 9.57E-05                                                    | 1.76E-07                                        | 10.97777        | 0.29773         | 0.511608           |
> >
> >    Table 2: The $\hat{y}$ is the actual model output, the $\hat{y}^{\prime}$ is the optimized model output, the $f$ is the actual feature map, and the $f^{\prime}$ is the obtained feature map.
> >
> > 4. Requirement on B to not exceed C+1: We have written the B<=C as a limitation of our method in the conclusion section in the rebuttal version.
> >
> > >**Q2  1. The technical contributions of this work are not particularly clear. The core techniques adopted by RGLA for enabling large batch recovery (e.g., disaggregating gradients and training a generator model) were discovered in prior work by Xue et al. and the inversion on the model output was discussed by Zhu & Blaschko et al. The only distinction seems to be the relaxation on duplicate labels. 2. Besides, some closely related works were not compared/discussed. For instance, [11] also trains a model to learn inverse mapping, and the idea of disaggregating and inverting feature vectors is very similar to the cocktail party attack [12].**
> >
> > 1. While it is true that Zhu & Blaschko et al. [9] have discussed the inversion of model outputs, their method is applicable only under very restricted conditions (B=1, C=2) and suffers from the challenge of twin data. In contrast, our RGLA approach formulates three innovative relations that enable the optimization of model outputs without the limitation of C. Our method accurately reconstructs model outputs as long as B<C, and it cleverly avoids the presence of twin data. We believe this is a significant advancement because once the model's output is leaked from the gradient, all methods in the model inversion attack domain (inverting the model's output to input) can be applied to gradient leakage attacks. The process of leaking the output and then the feature map to generate the original input, as described in our manuscript, is just one example. In reality, once the model's output is leaked, there are numerous methods in the model inversion attack domain that can be used to reconstruct the model's input. We believe this further emphasizes the uniqueness and importance of our contribution to this field.
> >
> > 2. We have included a discussion of related works [11, 12] in the related works of our rebuttal version.
> >
> > >**Q3 The proof of Proposition 2 made assumptions about the expressivity of the model, which should be made explicit in the text.**
> >
> > Thank you for your meticulous review of our proof. We explicitly state the assumptions about the expressibility of the model in Section 3 in the rebuttal version, which is the concern of you and the external comment from Dimitar Iliev Dimitrov.
> >
> > >**Q4 The image similarity-based privacy metrics have some inherent limitations. For instance, the reconstructed images on CelebA have high measured PSNR but barely reveal any practically identifiable information. It would be better to add corresponding discussions.**
> >
> > Thank you for pointing out this. We have discussed the limitations of the widely-used metrics in our rebuttal version.

---

> > > ### Author Response · Authors · 2023-11-18
> > > **Response to reviewer nX92 (Part 3)**
> > >
> > > >**Q5 1. From what the reviewer understands, the performance degradation of exiting attacks in the duplicated label scenario is an artifact of the label ambiguity. If that’s the case, existing optimization-based attacks should perform as well as if there were no duplicated labels in the extreme case where all images come from the same class. 2. That’s not what’s observed in Table 8 - existing attacks still perform poorly even if there is no label ambiguity. What causes the performance of existing attacks to drop? 3. How would these methods perform if the true labels are assumed to be known?**
> > >
> > > 1. Thank you for your insightful observations. The performance degradation of attacks in scenarios with duplicated labels is due to the entanglement of gradients corresponding to data with the same labels in the fully connected layer's gradient matrix, a phenomenon also described in previous work [13]. Since gradients are calculated through backpropagation, the gradients of the convolutional layers affected by this part also become entangled. As a result, the more duplicated labels there are, the more complex the entanglement of gradients becomes, making the optimization increasingly challenging. Analytical-based methods cannot reconstruct duplicated labels; thus, as the proportion of duplicated label data in a batch increases, the overall reconstruction metrics of the batch deteriorate.
> > >
> > > 2. The subpar performance of these methods in non-duplicated label scenarios is due to large batch size of 8 and high resolution of 224$\times$224. This results in a search space of 8$\times$224$\times$224 variables, far exceeding the maximum variable size these methods claim to reconstruct in their respective papers (as shown in Table 1 of our manuscript). Therefore, even in the absence of duplicated labels, their methods perform poorly.
> > > 3. In our comparative experiments, DLG, IG, and FGLA all used known true labels, while the GGL method used labels inferred by iDLG. For fairness in comparison, as also mentioned by reviewer iafz, we re-performed the GGL method with known labels and updated our experimental results in the rebuttal version. These results demonstrate that our proposed RGLA method still outperforms others in known ground-truth labels scenario.
> > >
> > > >**Q6 What learning task is considered for experiments on the CelebA dataset? How many classes are there? What is the auxiliary dataset used?**
> > >
> > > Thank you for your questions regarding our experiments on the CelebA dataset. The learning task conducted on CelebA was facial classification, encompassing a total of 10,177 classes. We have made this explicitly clear in the CelebA experiment in rebuttal version of our manuscript. We take ImageNet dataset as the auxiliary dataset, as we mentioned in our manuscript. To clarify this, we have revised our manuscript to state more explicitly that the default auxiliary dataset used in our experiments is ImageNet.
> > >
> > > >**Q7 The elimination of the twin solution seems to rely on its smaller loss value (as empirically verified in Fig. 3). How are these twin data eliminated on a well-trained model where all training data have relatively small loss values?**
> > >
> > > Thank you for your question regarding the elimination of twin solutions. In fact, the root cause of the emergence of twin solutions lies in the non-monotonicity of $\frac{\partial l(\hat{y}, y)}{\partial y}$. The loss value is just an experimental manifestation of the difference between the twin solution and the positive solution, when the model is well-trained, both the twin solution and the positive solution may have small loss values. In such cases, it becomes necessary to find other method to distinguish between twin solutions and correct solutions. We take this as our future work.
> > >
> > > In the rebuttal version of our manuscript, we have clarified in the threat model section that our attack primarily occurs during the model's initialization or pre-training phase. Regardless of the stage at which the attack takes place, any leakage of data is a significant security concern that warrants attention.

---

> > > > ### Author Response · Authors · 2023-11-18
> > > > **Response to reviewer nX92 (Part 4)**
> > > >
> > > > >**Q8 As RGLA relies on inverting the model output to feature maps, how would defense methods that perturb/modify the FCL part of the model (e.g., [13] [14]) affect the attack performance?**
> > > >
> > > > Thank you for your question. In response to the defense method in [13], it defends by perturbing the feature map. The training of the global model is affected when the feature map is largely perturbed, while our method is still effective in recovering the feature map when the perturbation is small. In order to verify the effectiveness of our method against the defense proposed in [13], we conducted experiments with the maximum defense parameter value (80%) set in the [1, 13] paper, and the results in Table 3 shows that our method is still effective.
> > > >
> > > > |                            | PSNR$\uparrow$ | SSIM$\uparrow$ | LPIPS$\downarrow$ |
> > > > | -------------------------- | -------------- | -------------- | ----------------- |
> > > > | RGLA without defense       | 18.79270       | 0.50706        | 0.55866           |
> > > > | RGLA with 80% Soteria [13] | 16.34467       | 0.44887        | 0.58806           |
> > > >
> > > > Table 3
> > > >
> > > > For the defense method mentioned in [14], it defends by embedding a variational module consisting of an encoder and a decoder in the network. Since this variational module contains randomly sampled vectors, it becomes very difficult to train a generator that is able to reverse from the feature map back to the original input, even if the feature map is accurate. Therefore, this defense method mainly defends our attack by affecting the training of the generator.
> > > >
> > > > >**Q9 It is interesting to see that gradient clipping and additive noise have little effect on FGLA as combining these two essentially provides some notion of differential privacy. What is the largest epsilon value that is able to defend FGLA and the proposed RGLA attack?**
> > > >
> > > > Thank you for your interesting question. Regarding gradient clipping defense, in our experiments, we utilized our auxiliary dataset to estimate the second norm of the gradients and used this information to restore clipped gradients, effectively mitigating the clipping defense. The implementation details are available in the source code provided in our supplementary materials. Simultaneously, FGLA extracts feature maps using the following equation, thereby directly bypassing the Clip defense. Therefore, we did not identify an effective largest epsilon value that could defend against both RGLA and FGLA.
> > > >
> > > > $f^\prime\approx\frac{{\nabla\theta^W_L}^\prime}{{\nabla\theta^b_L}^\prime}=\frac{\nabla\theta^W_L\cdot\frac{\mathcal{C}}{\|\nabla\theta\|}}{\nabla\theta^b_L\cdot\frac{\mathcal{C}}{\|\nabla\theta\|}}$
> > > >
> > > > As for the noisy gradient defense, we evaluate RGLA and FGLA under increasing noise levels. Table 4 shows the PSNR values of reconstructed results of RGLA and FGLA under different noise level. The experimental results are shown in the table below. If we consider a PSNR of 14 as the threshold, then the largest epsilon value that RGLA can reconstruct is 0.1, while for FGLA it is 0.06. We hope these experimental results answer your question and demonstrate the effectiveness of our methods.
> > > >
> > > > |            | 0.01  | 0.02  | 0.03  | 0.04  | 0.05  | 0.06  | 0.07  | 0.08  | 0.09  | 0.1   | 0.11  |
> > > > | ---------- | :---: | :---: | :---: | :---: | :---: | :---: | ----- | :---: | :---: | ----- | ----- |
> > > > | RGLA(Ours) | 18.90 | 18.97 | 18.41 | 17.76 | 17.08 | 16.43 | 15.81 | 15.23 | 14.72 | 14.25 | 13.83 |
> > > > | FGLA       | 18.83 | 18.79 | 18.08 | 17.23 | 15.87 | 14.19 | 12.92 | 11.43 | 10.24 | 9.78  | 9.39  |
> > > >
> > > > Table 4

---

> > ### Public Comment · ~Dimitar_Iliev_Dimitrov2 · 2023-11-19
> > **Re Q1**
> >
> > **Response to Q1:**
> > - Can the authors provide more information about the experiment in Table 1? In particular, what was the batch size and number of repeated class labels?
> > - Can the authors explore the connection between their auxiliary dataset and the one that they attack more thoroughly? That is, can they show how different level of data shifts will affect their results?

---

> > ### Comment · Reviewer_nX92 · 2023-11-19
> >
> > Thanks for the detailed response. Could you comment on why the LPIPS score is lower with the average pooling layer than without it? Could you provide some visual comparisons? Also a minor comment, I believe the "epsilon" in your response to Q9 refers to the variance of the noise rather than the privacy parameter?

---

> > > ### Author Response · Authors · 2023-11-20
> > > **Regarding the "lower LPIPS" for with average pooling layer**
> > >
> > > Thank you for your time and effort in reviewing our rebuttal. We hope the following response addresses your concerns.
> > >
> > > >**Q10 Could you comment on why the LPIPS score is lower with the average pooling layer than without it? Could you provide some visual comparisons?**
> > >
> > > LPIPS evaluates the similarity of images in the feature space after being transformed by a deep learning model. This metric does not directly compare pixel points but rather the feature representations of two images as transformed by a model, such as a VGG network. If these feature representations are close in the high-dimensional space, the LPIPS value will be low, even if the images differ significantly from a pixel perspective. Hence, the two images may appear visually distinct but could possess similarities at the feature level, leading to a low LPIPS score. On the other hand, when training our generator for networks with an average pooling layer, our loss function was -PSNR - 10×SSIM + 10×LPIPS. Due to the significant reduction of original data information in the feature maps by the pooling layer, the generator is unable to reconstruct all the details of the original image at the pixel level. However, to achieve a smaller loss value, the generator learns feature representations within its capability to lower the LPIPS score and, consequently, the overall loss. Therefore, when feature maps are input into the generator, it produces images that are visually dissimilar to the original images but have lower LPIPS values. To validate our intuition, we modified the loss function during generator training to -PSNR - 10 × SSIM (removing LPIPS) and conducted data attack experiments using this trained generator. The experiment results combined with those from the previous Table 2 are shown in Table 5. It is noteworthy that the three rows in Table 5 represent experiments conducted on the same 100 batches (with the same seed). From Table 5, it is evident that the generator, when using the loss function -PSNR - 10×SSIM + 10×LPIPS, indeed learned to generate images that are feature-wise similar to the original images, hence exhibiting lower LPIPS values. However, when using the loss function -PSNR - 10×SSIM, the generator no longer focuses on learning such features, resulting in a higher LPIPS. For the second and third rows, due to the reduced information contained in the feature maps by the pool layer and the limited capability of the generator, the original images could not be reconstructed at the pixel level. Consequently, this led to lower PSNR and SSIM values. We provide a visual comparison of the three cases in the image "with_without_pool.png" in the Supplementary Material.
> > >
> > > |                                                              | $\left\|\hat{y}^{\prime} - \hat{y} \right\|^2$ $\downarrow$ | $\left\|f^{\prime} - f \right\|^2$ $\downarrow$ | PSNR $\uparrow$ | SSIM $\uparrow$ | LPIPS $\downarrow$ |
> > > | ------------------------------------------------------------ | ----------------------------------------------------------- | ----------------------------------------------- | --------------- | --------------- | ------------------ |
> > > | Without average pooling layer                                | 2.00E-04                                                    | 1.01E-04                                        | 18.72215        | 0.48819         | 0.57847            |
> > > | With average pooling layer (loss = - PSNR - 10 $\times$ SSIM + 10$\times$LPIPS) | 9.57E-05                                                    | 1.76E-07                                        | 10.97777        | 0.29773         | 0.511608           |
> > > | With average pooling layer (loss = - PSNR - 10 $\times$ SSIM) | 9.57E-05                                                    | 1.76E-07                                        | 11.00478        | 0.333081        | 0.914162           |
> > >
> > > Table 5

---

> > > > ### Author Response · Authors · 2023-11-20
> > > > **Regarding the "epsilon"**
> > > >
> > > > >**Q11 Also a minor comment, I believe the "epsilon" in your response to Q9 refers to the variance of the noise rather than the privacy parameter?**
> > > >
> > > > Apologies for the confusion earlier, where we mistakenly interpreted "epsilon" as the variance of Gaussian noise. Now, we will address your question Q9 again. Based on our knowledge, when gradients are first clipped to a threshold value $\mathcal{C}$, and then Gaussian noise with a mean of 0 and variance $\sigma^2$ is added, "epsilon" can be calculated using the following formula: $\epsilon = \frac{\mathcal{C}}{\sigma} \sqrt{2 \ln(1.25 / \delta)}$
> > > >
> > > > where, $\mathcal{C}$ is the gradient clipping threshold, $\sigma$ is the standard deviation of the Gaussian noise, and $\delta$ is the minimum probability of privacy breach acceptable in differential privacy. In our experiments, we set $\delta$ to 0.1. A smaller value of $\epsilon$ means stronger privacy protection. We then varied the value of $\mathcal{C}$ from 16 to 2 and the variance of Gaussian noise $\sigma^2$ from 0.0001 to 0.1, and calculated the corresponding values of $\epsilon$ as shown in Table 6.
> > > >
> > > > Table 7 displays the reconstruction metric PSNR for the proposed RGLA method under the defense mechanisms combining different cropping $\mathcal{C}$ and noise variances $\sigma^2$ as shown in Table 6. The corresponding epsilon values can be referenced in Table 6. Table 8 presents the PSNR for the FGLA method under similar conditions. By synthesizing information from all three tables, it is apparent that the defense strength cannot be strictly measured by the size of epsilon alone, as the addition of noise has a greater impact on the proposed RGLA and FGLA methods than the cropping. For instance, both RGLA and FGLA perform better in the scenario with $\mathcal{C}=2$, $\sigma^2=0.001$, $\epsilon=142.14720$ than with $\mathcal{C}=8$, $\sigma^2=0.001$, $\epsilon=179.80357$.
> > > >
> > > >
> > > >
> > > > |    $\epsilon$    | $\sigma^2=0.0001$ | $\sigma^2=0.001$ | $\sigma^2=0.01$ | $\sigma^2=0.1$ |
> > > > | :--------------: | :---------------: | :--------------: | :-------------: | :------------: |
> > > > | $\mathcal{C}=16$ |    3596.07155     |    1137.17767    |    359.60715    |   113.71776    |
> > > > | $\mathcal{C}=8$  |    1798.03577     |    568.58883     |    179.80357    |    56.85888    |
> > > > | $\mathcal{C}=4$  |     899.01788     |    284.29441     |    89.90178     |    28.42944    |
> > > > | $\mathcal{C}=2$  |     449.50894     |    142.14720     |    44.95089     |    14.21472    |
> > > >
> > > > Table 6
> > > >
> > > >
> > > > |       PSNR       | $\sigma^2=0.0001$ | $\sigma^2=0.001$ | $\sigma^2=0.01$ | $\sigma^2=0.1$ |
> > > > | :--------------: | :---------------: | :--------------: | :-------------: | :------------: |
> > > > | $\mathcal{C}=16$ |     18.80273      |     18.76529     |    15.86617     |    7.62247     |
> > > > | $\mathcal{C}=8$  |     18.80224      |     18.63958     |    13.03881     |    6.71740     |
> > > > | $\mathcal{C}=4$  |     18.79842      |     18.13810     |    10.62324     |    6.45114     |
> > > > | $\mathcal{C}=2$  |     18.77972      |     16.64649     |     8.10522     |    6.35091     |
> > > >
> > > > Table 7: RGLA's performance under DP of varying $\epsilon$
> > > >
> > > >
> > > > |       PSNR       | $\sigma^2=0.0001$ | $\sigma^2=0.001$ | $\sigma^2=0.01$ | $\sigma^2=0.1$ |
> > > > | :--------------: | :---------------: | :--------------: | :-------------: | :------------: |
> > > > | $\mathcal{C}=16$ |     18.79882      |     18.73356     |    12.69809     |    7.90765     |
> > > > | $\mathcal{C}=8$  |     18.79714      |     18.53454     |     9.60690     |    7.88662     |
> > > > | $\mathcal{C}=4$  |     18.78956      |     17.78933     |     8.09459     |    7.90427     |
> > > > | $\mathcal{C}=2$  |     18.75774      |     14.63371     |     7.98085     |    7.74494     |
> > > >
> > > > Table 8: FGLA's performance under DP of varying $\epsilon$

---

> > > > > ### Comment · Reviewer_nX92 · 2023-11-20
> > > > >
> > > > > Thank the authors for your extensive clarification and additional experiments that have addressed most of my concerns. Despite having made some adversarial assumptions, I recognize the merits of this work in effectively tackling challenges related to large batch size and duplicated labels, demonstrating compelling empirical performance. I've adjusted my evaluation to support accepting this paper. I encourage the authors to expand the limitation sections in the final version to deepen the discussion on relevant aspects.

---

> > > > > > ### Author Response · Authors · 2023-11-21
> > > > > > **Thanks for the recognition**
> > > > > >
> > > > > > Thank you very much for your thoughtful and constructive feedback. We are grateful for your recognition of the merits of our work, especially in addressing challenges related to large batch sizes and duplicated labels, and we appreciate your support in accepting our paper. We take your advice very seriously and will definitely expand the limitations section in the final version of our manuscript. We believe that a thorough discussion of the limitations not only enhances the transparency of our research but also provides valuable insights for future work in this area. Your comments have been instrumental in improving the quality of our paper, and we thank you again for your valuable time and effort in reviewing our work.

---

> ### Author Response · Authors · 2023-11-18
> **Response to reviewer nX92 (Part 5)**
>
> Reference:
>
> [1] Jinwoo Jeon, Jaechang Kim, Kangwook Lee, Sewoong Oh, and Jungseul Ok. Gradient inversion with generative image prior. In Neural Information Processing Systems, pp. 29898–29908, 2021.
>
> [2] Dongyun Xue, Haomiao Yang, Mengyu Ge, Jingwei Li, Guowen Xu, and Hongwei Li. Fast generation-based gradient leakage attacks against highly compressed gradients. IEEE INFOCOM 2023 - IEEE Conference on Computer Communications, 2023.
>
> [3] Zhuohang Li, Jiaxin Zhang, Luyang Liu, and Jian Liu. Auditing privacy defenses in federated learning via generative gradient leakage. In Proceedings of the IEEE/CVF Conference on Computer Vision and Pattern Recognition, pp. 10132–10142, 2022.
>
> [4] Yang H, Ge M, Xiang K, et al. Using Highly Compressed Gradients in Federated Learning for Data Reconstruction Attacks[J]. IEEE Transactions on Information Forensics and Security, 2022, 18: 818-830.
>
> [5] Hanchi Ren, Jingjing Deng, and Xianghua Xie. Grnn: generative regression neural network—a data leakage attack for federated learning. ACM Transactions on Intelligent Systems and Technology (TIST), 13(4):1–24, 2022.
>
> [6] Yue K, Jin R, Wong C W, et al. Gradient obfuscation gives a false sense of security in federated learning[C]//32nd USENIX Security Symposium (USENIX Security 23). 2023: 6381-6398.
>
> [7] Wainakh A, Ventola F, Müßig T, et al. User-level label leakage from gradients in federated learning[J]. arXiv preprint arXiv:2105.09369, 2021.
>
> [8] Ma K, Sun Y, Cui J, et al. Instance-wise Batch Label Restoration via Gradients in Federated Learning[C]//The Eleventh International Conference on Learning Representations. 2022.
>
> [9] Zhu J, Blaschko M. R-gap: Recursive gradient attack on privacy[J]. arXiv preprint arXiv:2010.07733, 2020.
>
> [10] Jonas Geiping, Hartmut Bauermeister, Hannah Droge, and Michael Moeller. Inverting gradients - ¨how easy is it to break privacy in federated learning? ArXiv, abs/2003.14053, 2020.
>
> [11] Wu, Ruihan, et al. "Learning to invert: Simple adaptive attacks for gradient inversion in federated learning." Uncertainty in Artificial Intelligence. PMLR, 2023.
>
> [12] Kariyappa, Sanjay, et al. "Cocktail party attack: Breaking aggregation-based privacy in federated learning using independent component analysis." International Conference on Machine Learning. PMLR, 2023.
>
> [13] Sun, Jingwei, et al. "Soteria: Provable defense against privacy leakage in federated learning from representation perspective." Proceedings of the IEEE/CVF conference on computer vision and pattern recognition. 2021.
>
> [14] Scheliga, Daniel, Patrick Mäder, and Marco Seeland. "Precode-a generic model extension to prevent deep gradient leakage." Proceedings of the IEEE/CVF Winter Conference on Applications of Computer Vision. 2022.

---

> ### Author Response · Authors · 2023-11-22
> **Provide more information about the experiment in Table 1 and discussion on how the similarity between the auxiliary dataset and the attacked one affect the reconstruction results**
>
> >**Q1 Can the authors provide more information about the experiment in Table 1? In particular, what was the batch size and number of repeated class labels?**
>
> The experiments setup for Table 1 consists of the attacked dataset: cifar100, the attacked network: resnet50, the batch size: 8, and the labels distribution of is assigned randomly. We think you want to see how the combination of iLRG and RGLA performs with repeated labels, so we added the experiment in the case of 4 repeated labels and a batch size of 8, and merged the results with Table 1, as shown in Table 5. From Table 5 we can see that repeated labels have no effect on the combination of iLRG with RGLA, which further validates the reasonableness of label assumptions.
>
> |                                                            | PSNR$\uparrow$ | SSIM$\uparrow$ | LPIPS$\downarrow$ |
> | ---------------------------------------------------------- | -------------- | -------------- | ----------------- |
> | Ground-truth label + RGLA                                  | 19.24153       | 0.50947        | 0.55870           |
> | iLRG [1] + RGLA (Randomly assigned label distribution)      | 19.22677       | 0.49973        | 0.56795           |
> | iLRG [1] + RGLA (Duplicate 4 labels within batch size of 8) | 19.16298       | 0.48339       | 0.57021          |
>
> Table 5
>
> [1] Ma K, Sun Y, Cui J, et al. Instance-wise Batch Label Restoration via Gradients in Federated Learning[C]//The Eleventh International Conference on Learning Representations. 2022.
>
>
> >**Q2 Can the authors explore the connection between their auxiliary dataset and the one that they attack more thoroughly? That is, can they show how different level of data shifts will affect their results?**
>
> Our auxiliary dataset is ImageNet and experiments have been performed on ImageNet, CelebA, Cifar10, and Cifar100, our intuition is that the similarity of the auxiliary dataset to the attacked dataset as well as the complexity of the auxiliary dataset to the attacked dataset affect the results.
>
> - Similarity between the auxiliary dataset and the attacked dataset affect the result. The similarity between ImageNet and the Cifar100 dataset is greater than that between ImageNet and the CelebA dataset, and from the experiment results on the right of Fig. 2 in the manuscript we can see that the reconstruction results on the cifar100 dataset are better than those on CelebA. Therefore we draw this conclusion the more similar the auxiliary dataset and the attacked dataset are the more favourable to the attack. However, there is no good way to quantify this similarity between the two datasets, so for the time being, we do not have a good way to quantitatively state how much the difference between the auxiliary dataset and the attacked dataset affects the results. However, we feel that the attacker can adjust the auxiliary dataset when performing reconstruction attacks. For example, if the attacker originally used ImageNet to attack CelebA, and when the reconstructed result can be visually seen as a human face, but may not be clearly identifiable, we can connect some face images to feed into the generator before performing the attack, and the result may become better. In other words, we can adjust the auxiliary dataset of our generator according to the reconstructed result.
> - The complexity of the auxiliary dataset being greater than that of the attacked dataset is advantageous for attacks. For instance, using ImageNet as an auxiliary dataset to attack Cifar100 allows the generator to learn more complex details, which is beneficial for the attack.

---

> ### Public Comment · ~Dimitar_Iliev_Dimitrov2 · 2023-11-22
> **Response to Authors**
>
> **Q1**: Thank you for this. I believe including an even larger level of label duplication - e.g something like batch size 64 with several sets of repeated labels will be better suited for the final manuscript. That said, I am really grateful for the provided extended results.
>
>  **Q2**: I think it will be nice if the authors include this discussion somewhere in their appendix in the next revision.

---

### Official Review · Reviewer_p8x8 · 2023-11-06

**Soundness:** 4 excellent
**Presentation:** 3 good
**Contribution:** 3 good
**Rating:** 8
**Confidence:** 4

**Summary:**

This paper proposes a novel gradient leakage attack RGLA, which first invert loss and FC layer gradients to get the final feature map before FC layer, then use a generator to map the final feature maps back to input space to get the reconstructed inputs. The method is shown effective to reconstruct high-resolution (224 x 224) pixels with large batch size (256) with duplicated labels.

**Strengths:**

1. The approach is thoughtfully designed, beginning with a solid theoretical analysis that logically leads to its expected effectiveness.

2. The experiments conducted in the paper are extensive and thorough. The authors have performed extensive ablation studies to show the performance. The results are impressive, showcasing good reconstruction fidelity and computational efficiency, thus highlighting the method's effectiveness.

3. The paper is well-written, presenting its concepts and findings with clarity and precision, leaving no ambiguity.

**Weaknesses:**

1. The paper lacks sufficient details regarding the training of the generator responsible for mapping final feature maps back into the original input samples. As readers, we are left with questions about the level of effort required to train such a generator and the upper bounds of its capabilities. Additionally, it would be valuable to understand how the performance of the feature inverter generator is affected by the increasing accuracy of the victim model. Could you provide more information into these aspects to enhance our understanding of your work?

2. A small weakness: I notice in the code provided in supplementary materials, the authors made some change to the architecture of Resnet - They remove the average pooling layer (over a 7x7 feature map), thus by stretching the final feature map, the FC layer is indeed 49 times wider than before. This inevitably simplifies the difficulty of reconstructions a lot.  To me this is not a severe concern, since the major part of paper does not assume how feature maps are reshaped to enter the FC layers, and this technical modification is not the focus of the research problem.  But for the researchers studying this problem, they could be concerned about this.  I wonder could authors show how the results will be like, if not removing average pooling, and use the original Resnet architecture in Torchvision implementation? It is expected to see performance drop in that case, but revealing this could give readers more insights about the difficulty of the problem, and understand how much information will be lost by average pooling before FC layer.

**Questions:**

1. My primary question is about the information available to the attacker through gradients, which is essentially an aggregated representation from a batch of inputs. I'm curious about how your method manages to distinguish features of different images within the same class. I couldn't identify explicit constraints in the optimization objective that encourage disentanglement between samples from the same class. Intuitively, I would expect that reconstructed samples from the same class might exhibit entangled features, resulting in averaged reconstruction or mixed semantics. Could you provide additional insights, either theoretical or empirical, to clarify how your approach achieves this distinction? This would be my biggest concern for understanding how your approach works.

2. It appears that the reconstructed samples exhibit a degree of over-smoothing, lacking sharp edges and finer details. This effect seems reminiscent of applying a large Total Variation loss or L2 loss. Could you please explain the reasons behind this observation? Is it related to the characteristics of the feature inversion generator? Are there potential room for improvement? If the level of smoothness can be adjusted, what will the results look like if they do not appear so smooth?

3. See weakness 2, I am curious about the performance of proposed approach when using average pooling layer before the final FC layer.

---

> ### Author Response · Authors · 2023-11-18
> **Response to reviewer p8x8 (Part 1)**
>
> Thank you very much for your praise on our manuscripts, such as "innovative attacks", "insightful design", "solid theoretical analyses", "extensive and thorough experiments", "well-written", and so on. We are honored to receive such praise and we do our best to address your concerns and questions.
>
> >**Q1  1. There is a lack of detailed information on the training of the generator used for mapping final feature maps to original input samples and clarity on the effort required to train this generator. 2. How the generator's performance is influenced by the increasing accuracy of the victim model.**
>
> Thank you very much for your question.
>
> 1. The structure of the generator and its training details have been thoroughly described in the prior work [1]. Here, I would like to add some details that were not mentioned in [1]: the generator has a total of 100,590,979 parameters, and consumes about 23GB of video memory when the batch size is 32 during training. When experimenting on ImageNet, it takes about 2 hours to complete an epoch, and typically after about 4 epochs of training, the generator is able to effectively invert the feature map back to the original model input. This generator training does not affect the efficiency of our attacks. This is because it is becoming increasingly common to use a pre-trained convolutional layer plus an untrained fully-connected layer and to fix the convolutional layer to train only the fully-connected layer during the training process. In this case, we only need to train one generator to attack the private data of multiple clients in multiple rounds of federated learning.
>
> 2. As the accuracy of the attacked model increases, the feature map will contain more information about the original data, potentially reducing the time required to train an efficient generator.

---

> > ### Comment · Reviewer_p8x8 · 2023-11-19
> > **Regarding "pre-trained convolutional layer and untrained FC layer"**
> >
> > "it is becoming increasingly common to use a pre-trained convolutional layer plus an untrained fully-connected layer and to fix the convolutional layer to train only the fully-connected layer during the training process."
> >
> > Do you mean that a such a pre-trained feature inversion generator is trained, each time the attacked model changes its parameters, attacker only needs retraining on FC layers to adapt to the new weight?  Is it always the case? In your experiment setting, do you assume the victim model is trained to a certain state, if it is, then how many epochs have been trained on the victim model?

---

> > > ### Author Response · Authors · 2023-11-20
> > > **Thanks for reviewing our rebuttal**
> > >
> > > Thank you for your time and effort on reviewing our rebuttal. The statement "it is becoming increasingly common to use a pre-trained convolutional layer plus an untrained fully-connected layer and to fix the convolutional layer to train only the fully-connected layer during the training process" refers to the training mode of the attacked model, which often uses pre-trained convolutional layers with untrained fully-connected layers. Since the generator is used in our method to invert the convolutional layers of the attacked network, let us discuss the changes in the parameters of the convolutional layers during the training of the attacked model. If the attacked model fixes the convolutional layers and only trains the fully-connected layers during its training process (which is a possible scenario), then the attacker would not need to retrain the generator, as the convolutional layer parameters do not change. This situation is most favorable for the attacker. However, if the convolutional layer parameters of the attacked model are involved in the training, their changes after each iteration would be minimal due to the prior pre-training. In this case, only minor adjustments (fine-tuning) to the generator would be necessary to maintain its capacity to invert the convolutional layers. Therefore, in scenarios using pre-trained convolutions, there is no need to train the generator from scratch in each iteration.
> > >
> > > Yes, in our experimental setup, we used a pre-trained ResNet50 model provided by PyTorch, which has already been trained to a certain state.

---

> ### Author Response · Authors · 2023-11-18
> **Response to reviewer p8x8 (Part 2)**
>
> >**Q2 A small weakness: I notice in the code provided in supplementary materials, the authors made some change to the architecture of Resnet - They remove the average pooling layer (over a 7x7 feature map), thus by stretching the final feature map, the FC layer is indeed 49 times wider than before. This inevitably simplifies the difficulty of reconstructions a lot. To me this is not a severe concern, since the major part of paper does not assume how feature maps are reshaped to enter the FC layers, and this technical modification is not the focus of the research problem. But for the researchers studying this problem, they could be concerned about this. I wonder could authors show how the results will be like, if not removing average pooling, and use the original Resnet architecture in Torchvision implementation? It is expected to see performance drop in that case, but revealing this could give readers more insights about the difficulty of the problem, and understand how much information will be lost by average pooling before FC layer.**
>
> Thank you for meticulously reviewing our code and providing valuable comments. Indeed, we removed the average pooling layer over the fully-connected layer (FC layer) in our experiments in order to make the feature map larger so that it contains more information about the original data, which is consistent with what was done in [1]. In order to explore how much information is lost by the average pooling layer, we conducted experiments in two scenarios: one in which the pooling layer is removed and the other in which the pooling layer is retained. And keep the rest of the experimental settings the same, including a batch size of 8, a dataset of ImageNet, and attacking the same 100 data batches. According to the experimental results in Table 1, whether the pooling layer is removed or not, our proposed method is able to reconstruct model outputs that are very close to the actual model outputs, which further reconstructs feature maps that are similar to the original feature maps. However, in the setting with the pool layer, the generator can not generate the original data even with the precise feature map. This phenomenon can be attributed to the fact that the addition of the pooling layer reduces the information about the original data contained in the feature maps, making it difficult for the generator to reconstruct the original image from feature maps that contain less information about the original data. The contribution of our manuscript does not lie in the training of the generator, which is the main contribution of [1]. In the future, we also expect more powerful generator models to be able to reconstruct the original images from corresponding feature maps even with limited information contained.
>
> |                               | $\left\|\hat{y}^{\prime} - \hat{y} \right\|^2$ $\downarrow$ | $\left\|f^{\prime} - f \right\|^2$ $\downarrow$ | PSNR $\uparrow$ | SSIM $\uparrow$ | LPIPS $\downarrow$ |
> | ----------------------------- | ----------------------------------------------------------- | ----------------------------------------------- | --------------- | --------------- | ------------------ |
> | Without average pooling layer | 2.00E-04                                                    | 1.01E-04                                        | 18.72215        | 0.48819         | 0.57847            |
> | With average pooling layer    | 9.57E-05                                                    | 1.76E-07                                        | 10.97777        | 0.29773         | 0.511608           |
>
> Table 1: The $\hat{y}$ is the actual model output, the $\hat{y}^{\prime}$ is the optimized model output, the $f$ is the actual feature map, and the $f^{\prime}$ is the obtained feature map.

---

> > ### Author Response · Authors · 2023-11-18
> > **Response to reviewer p8x8 (Part 3)**
> >
> > >**Q3 My primary question is about the information available to the attacker through gradients, which is essentially an aggregated representation from a batch of inputs. I'm curious about how your method manages to distinguish features of different images within the same class. I couldn't identify explicit constraints in the optimization objective that encourage disentanglement between samples from the same class. Intuitively, I would expect that reconstructed samples from the same class might exhibit entangled features, resulting in averaged reconstruction or mixed semantics. Could you provide additional insights, either theoretical or empirical, to clarify how your approach achieves this distinction? This would be my biggest concern for understanding how your approach works.**
> >
> > This is a good question. Our reconstructed images are generated from feature maps by a generator. Therefore, the problem of distinguishing different images within the same class becomes a problem of distinguishing their feature maps. According to Equation (5) in our paper, the computation of feature maps is as follows: $\boldsymbol{f}^{\prime}=\left(\frac{\partial l\left(\hat{\boldsymbol{y}}^{\prime},\boldsymbol{y}\right)}{\partial\hat{\boldsymbol{y}}^{\prime}}\right)^{-1}\cdot\nabla\boldsymbol{\theta}_L^W$
> >
> > With known labels and gradients, the feature maps are determined by the optimized model outputs. Our research actually addresses how to distinguish model outputs of different images within the same class. The results of the model output depend on our optimization objective:
> >
> > $\mathcal{L}_w = \left\|\nabla\boldsymbol{\theta}_L^W\cdot\boldsymbol{\theta}_L^W-\frac{\partial l\left(\hat{\boldsymbol{y}}^{\prime},\boldsymbol{y}\right)}{\partial\boldsymbol{\hat{\boldsymbol{y}}}^{\prime}}\cdot\left(\hat{\boldsymbol{y}}^{\prime},-\boldsymbol{\theta}_L^b\right)\right\|^2$,
> >
> > $\mathcal{L}_ b = \left\|\nabla\boldsymbol{\theta}_{L}^{b}-\frac{\partial l(\hat{\boldsymbol{y}}^{\prime},\boldsymbol{y})}{\partial\hat{\boldsymbol{y}}^{\prime}}\cdot\mathbb{I}\right\|^{2}$
> >
> > $\mathcal{L}_{loss} = \left\|l(\hat{\boldsymbol{y}}^{\prime},\boldsymbol{y})-[-\frac1B\sum _{i=1}^B\ln(\nabla\boldsymbol{\theta} _{L\boldsymbol{y}(i)}^b+\frac{\lambda _{\boldsymbol{y}(i)}}B)]\right\|^2$
> >
> > In our loss objective, the model output considers all samples in a batch, with a size of B$\times$C, rather than an average output. $\mathcal{L}_ w$ establishes equations for every element in the model output, $\mathcal{L}_ {b}$ establishes equations for model outputs within the same class, and $\mathcal{L}_ {loss}$ establishes the overall equation for the entire model output. Therefore, an averaged reconstructed model output might satisfy $\mathcal{L}_ b$ and $\mathcal{L}_ {loss}$, but not satisfy $\mathcal{L}_ w$, as $\mathcal{L}_ w$ requires every element in the model output to meet the equation without any summing or averaging. Thus, our method does not result in similar model outputs for data in the same class, thereby not reconstructing averaged images for the original images within the same class.
> >
> > The key difference between our method and previous ones is that the latter mainly match averaged gradients, while our method establishes equations for every element in the model output, providing a more granular approach. Only when every element of the model output satisfies the optimization objective will the optimization loss reach its minimum value. Therefore, our method can distinguish images within the same class.

---

> > > ### Comment · Reviewer_p8x8 · 2023-11-19
> > > **Thanks for the response**
> > >
> > > Many thanks for your comprehensive and insightful response! Your provided information and insights have effectively addressed most of my concerns. I also took the time to go through other reviews and the author's rebuttal, from which I learned some interesting points. In my assessment, the paper demonstrates novelty and is supported by well-executed experiments that show satisfying improvements in results. Furthermore, the authors' transparency regarding technical details are valuable. As a result, I would like to raise my score from 6 to 8.

---

> > > > ### Author Response · Authors · 2023-11-20
> > > > **Thanks for raising the score**
> > > >
> > > > We sincerely appreciate your kind words and are grateful for your acknowledgment of the efforts we have put into addressing the concerns. We thank you a lot for your time in reviewing other feedback and rebuttals, finding them informative and useful. It is encouraging to hear that our paper's novelty and the rigor of our experimental work have resonated well with you, and we are especially pleased that the improvements and transparency in our technical approach have been positively received.
> > > >
> > > > Your decision to raise the score from 6 to 8 is greatly appreciated and motivates us to continue striving for excellence in our research endeavors. We will remain committed to maintaining this level of detail and clarity in our work and look forward to contributing further to the field.
> > > >
> > > > Once again, thank you for your constructive feedback and support. Your insights have been invaluable in improving our work, and we are grateful for the opportunity to refine our research through this interactive review process.

---

> ### Author Response · Authors · 2023-11-18
> **Response to reviewer p8x8 (Part 4)**
>
> >**Q4 It appears that the reconstructed samples exhibit a degree of over-smoothing, lacking sharp edges and finer details. This effect seems reminiscent of applying a large Total Variation loss or L2 loss. Could you please explain the reasons behind this observation? Is it related to the characteristics of the feature inversion generator? Are there potential room for improvement? If the level of smoothness can be adjusted, what will the results look like if they do not appear so smooth?**
>
> Thank you for your question! Regarding the excessive smoothing that occurs in the reconstructed samples, as well as the lack of sharp edges and fine details, this may indeed be related to the Total Variation loss or L2 loss that we used. In addition to this, we believe it is also related to the architectural characteristics of our generator. Our generator employs a multi-layer convolutional and deconvolutional network, complemented by LeakyReLU activation functions and batch normalization (BatchNorm). This design helps produce smoother images and mitigating overfitting to some extent. However, it might also lead to the loss of high-frequency details in images. During the training of the generator, we used Mean Squared Error Loss (MSELoss) along with image quality metrics such as PSNR, SSIM, and LPIPS. The combination of these loss functions might further enhance the smoothing effect, particularly as PSNR and SSIM tend to assess overall structural similarity rather than precise detail reproduction. Regarding improvements to reduce image over-smoothing, we believe that adjustments to the loss function, optimization of the generator’s network structure, or the incorporation of techniques specifically aimed at high-frequency detail restoration could yield enhancements. For instance, exploring more complex loss functions like Perceptual Loss or losses used in Generative Adversarial Networks (GANs) might encourage the generation of more detailed and textured images. However, due to time constraints during the rebuttal phase, we were unable to explore more optimization functions and more robust generator models. We hope the reviewers can understand us.
>
> >**Q5 See weakness 2, I am curious about the performance of the proposed approach when using the average pooling layer before the final FC layer.**
>
> See our answer to Q2.
>
> Reference:
>
> [1] Dongyun Xue, Haomiao Yang, Mengyu Ge, Jingwei Li, Guowen Xu, and Hongwei Li. Fast generation-based gradient leakage attacks against highly compressed gradients. IEEE INFOCOM 2023 - IEEE Conference on Computer Communications, 2023.

---

> > ### Public Comment · ~Dimitar_Iliev_Dimitrov2 · 2023-11-19
> > **Re: Q4**
> >
> > **Response to Q4:** The authors might want to try to use diffusion-based models [1] for their generator as they are SOTA for image generation. That said, I am curious if the authors know for certain that the smoothing effect is not caused by the loss of information within the pre-FC convolution feature map instead? To me, it makes sense that the CNN, even without pooling layers, will want to try to "forget" some details that are not relevant to the classification task.
> > [1] Florinel-Alin Croitoru, Vlad Hondru, Radu Tudor Ionescu, and Mubarak Shah. Diffusion models in vision: A survey. IEEE Transactions on Pattern Analysis and Machine Intelligence, 2023.

---

> > > ### Author Response · Authors · 2023-11-21
> > > **Thanks for your interest and attention again**
> > >
> > > Thank you for sharing your insights. We agree with your perspective and believe that the smoothness observed in the reconstructed images, resulting in the loss of some details, could be attributed to several factors:
> > >
> > > 1. The feature maps may not contain all the details of the original data.
> > > 2. The generator's capacity may be limited, restricting its ability to restore every detail.
> > > 3. The training setup for the generator might not be optimal, possibly leading to an inadequately trained generator.
> > >
> > > Do you think there are any other reasons that might contribute to this smoothness beyond the ones mentioned?

---

> ### Public Comment · ~Dimitar_Iliev_Dimitrov2 · 2023-11-21
> **Re: Thanks for your interest and attention again**
>
> I think this covers the list, yes. That said, works like [1,2] show that simple decoders directly on the gradients work ( I appreciate the setup here is slightly different but still ), so I am leaning towards point 1. I appreciate that the authors are busy with the rest of the rebuttal, so I won't be bothering them on this anymore, but maybe consider training a diffusion model after the deadline. Might be able to tell us to what extent the problem is based on 2 or 3.
>
> [1] https://openreview.net/forum?id=Gt_GiNkBhu&referrer=%5Bthe%20profile%20of%20Chuan%20Guo%5D(%2Fprofile%3Fid%3D~Chuan_Guo1)
> [2] https://arxiv.org/abs/2306.03013

---

> > ### Author Response · Authors · 2023-11-22
> >
> > Thanks for your understanding, we're sure it'll be an interesting exploration, and right now we're leaning toward the first reason as well.

---

### Public Comment · ~Dimitar_Iliev_Dimitrov2 · 2023-11-10
**Public Comment**

Dear authors,

I have a few small comments/questions/suggestions for this submission. I hope you take them into account for the next version of the paper. Further, I hope the reviewers do not mind my feedback.

- The proof of Proposition 2 in Appendix A.2. is a bit unclear around Eq. 14. In particular, I am not sure what are $i,j,k$, and $t$ there.
- In Table 5, in Appendix A.3, what is the L2 loss referring to? The final image reconstruction loss, $\mathcal{L}_{text{loss}}$, or something else?
- In [1], a ResNet with removed pooling layers is used as the attacked network. Do the authors assume the same setup? If so, can the authors clearly state in Section 3?
- In Appendix A.3, the authors say: “...our attack primarily occurs during the initialization or pre-training phase of the model training...”. It would be nice if this limitation is also explicitly stated in Section 3.
- On page 6 top, the authors suggest that the twin solutions come from the overdetermined system. I do not think this is correct. Instead, in [2], it is shown the reason is that the equation is non-linear ( due to $\frac{\partial l} {\partial \hat{y}’}$ being non-linear ). I think this is the reason here, as well.
- FGLA [1] is advertised as a highly effective attack against compressed gradients. [1] claims that FGLA can handle 99.2% of the gradients being compressed. Do I understand correctly that the proposed method improves on this by allowing reconstruction from 99.9% of the gradients? Can you provide in the appendix a full comparison between the two methods on different levels of gradient compression?
- In Section 5.3 you cite the wrong paper for the choice of $\mathcal{C} = 4$. Should be [3]

[1] Dongyun Xue, Haomiao Yang, Mengyu Ge, Jingwei Li, Guowen Xu, and Hongwei Li. Fast generation-based gradient leakage attacks against highly compressed gradients. IEEE INFOCOM 2023 - IEEE Conference on Computer Communications, 2023.
[2] Junyi Zhu and Matthew Blaschko. R-gap: Recursive gradient attack on privacy. arXiv preprint arXiv:2010.07733, 2020.
[3] Abadi, Martin, et al. "Deep learning with differential privacy." Proceedings of the 2016 ACM SIGSAC conference on computer and communications security. 2016.

---

> ### Author Response · Authors · 2023-11-18
> **Thanks for your attention and suggestions**
>
> It is a great honor to receive your attention on our manuscript. These comments you raise are quite professional and will improve the quality of our paper. Here are our answers to your questions:
>
> >**The proof of Proposition 2 in Appendix A.2. is a bit unclear around Eq. 14. In particular, I am not sure what are $i$,$j$,$k$, and $t$ there.**
>
> In equation (14), $i$ and $k$ denote the indexes of the samples within the input batch; $j$ and $t$ denote the indexes of the class. We have added the above explanation near equation (14) in the rebuttal version.
>
> >**In Table 5, in Appendix A.3, what is the L2 loss referring to? The final image reconstruction loss, $\mathcal{L}_{textloss}$, or something else?**
>
> In Table 5, the L2 loss refers to the squared difference between the actual model's cross-entropy loss and the estimated model's cross-entropy loss, which is our loss term $\mathcal{L}_{loss}$
>
> >**In [1], a ResNet with removed pooling layers is used as the attacked network. Do the authors assume the same setup? If so, can the authors clearly state in Section 3?**
>
> Yes, we have made the same assumption about removing the pooling layers as in [1], and we have clearly stated this in our manuscript.
>
> >**In Appendix A.3, the authors say: “...our attack primarily occurs during the initialization or pre-training phase of the model training...”. It would be nice if this limitation is also explicitly stated in Section 3.**
>
> We appreciate your suggestion and have added this point in Section 3 of our rebuttal version.
>
> >**On page 6 top, the authors suggest that the twin solutions come from the overdetermined system. I do not think this is correct. Instead, in [2], it is shown the reason is that the equation is non-linear ( due to $\frac{\partial l}{\partial\hat{y}^\prime}$ being non-linear ). I think this is the reason here, as well.**
>
> Thank you for pointing out this claim. The reason for the existence of twin solutions is indeed the non-linearity of $\frac{\partial l}{\partial\hat{y}^\prime}$, and we realized this after submitting our manuscripts. We have corrected this in our rebuttal version.
>
> >**FGLA [1] is advertised as a highly effective attack against compressed gradients. [1] claims that FGLA can handle 99.2% of the gradients being compressed. Do I understand correctly that the proposed method improves on this by allowing reconstruction from 99.9% of the gradients? Can you provide in the appendix a full comparison between the two methods on different levels of gradient compression?**
>
> Yes, our proposed method improves on this by allowing reconstruction from 99.9% of the gradients. We have provided a comprehensive comparison between FGLA [1] and RGLA under different levels of gradient compression in the appendix of our rebuttal version.
>
> >**In Section 5.3 you cite the wrong paper for the choice of $\mathcal{C}=4$. Should be [3]**
>
> We've corrected this reference in the rebuttal version.
>
> [1] Dongyun Xue, Haomiao Yang, Mengyu Ge, Jingwei Li, Guowen Xu, and Hongwei Li. Fast generation-based gradient leakage attacks against highly compressed gradients. IEEE INFOCOM 2023 - IEEE Conference on Computer Communications, 2023.
>
> [2] Junyi Zhu and Matthew Blaschko. R-gap: Recursive gradient attack on privacy. arXiv preprint arXiv:2010.07733, 2020.
>
> [3] Abadi, Martin, et al. "Deep learning with differential privacy." Proceedings of the 2016 ACM SIGSAC conference on computer and communications security. 2016.

---

> > ### Public Comment · ~Dimitar_Iliev_Dimitrov2 · 2023-11-19
> > **Response**
> >
> > Thank you for your responses. I think the authors have addressed my concerns.

---

### Meta-Review · Area_Chair_hKG2 · 2023-12-09

**Metareview:**

This paper proposes RGLA, a new gradient inversion attack that enables sample recovery from a large batch of samples with duplicated labels. The key insight is that the per-sample model output of a linear classification head can be recovered given the model weights, average gradient, and ground truth labels. Once the model outputs are obtained, it is straightforward to recover the per-sample embeddings, and then use feature inversion to recover the training samples. The authors evaluate their attack on ImageNet, CIFAR and CelebA against several baselines including IG, GGL and FGLA, shows that RGLA can achieve better inversion accuracy under duplicated labels and various defenses.

This is a very difficult paper to judge because the reviews are divergent (scores 3, 6, 8) and there are solid reasons for both acceptance and rejection. AC ended up reading through the paper carefully to form an **independent opinion** and act as tie-breaker. In the AC's view, the attack recipe follows that of FGLA, and RGLA inherits the same strengths (e.g. large batch recovery, computational efficiency) and weaknesses (e.g. feature may not be invertible, reliance on a strong image prior to train the feature inversion model). The only technical contribution of the paper is defining the optimization objective in Eq. 7 that can successfully recover per-sample model outputs under duplicated labels. This can be a valuable enough contribution to warrant acceptance, but the current paper falls short due to the following reasons:
1. There is no additional improvement to the recipe of FGLA. Can the feature inversion model be trained on lower-dimensional feature vectors, even ones after pooling? Is it possible to circumvent the restriction of $B > C+1$ by using other priors?
2. Comparison against prior work is incomplete. Fig. 4 (left) is the only comparison with prior work under no defense. This is done using a fixed batch size, fixed dataset, and omitting some very relevant baselines such as STG [Yin et al., 2021], GI-GIP [Jeon et al., 2021] and Cocktail party attack [Kariyappa et al., 2023]. Overall, the ablation studies on RGLA seem thorough but baseline comparison is on the weaker side.
3. The experiment section is missing important details that affect the subjectivity of baseline comparisons. What is the dataset used in Fig. 4 (left)? Since the feature inversion model is trained on ImageNet, this dataset should be sufficiently different to ensure fairness. It also appears that the linear classification head is randomly initialized, which gives RGLA an unfair advantage since it explicitly relies on this assumption in Prop. 2. How are hyperparameters tuned for RGLA and baseline methods to ensure they are equally optimal?

AC strongly recommends the authors to take this feedback into consideration for the next revision.

**Justification For Why Not Higher Score:**

See reasons listed in the metareview.

**Justification For Why Not Lower Score:**

N/A

---

### Decision · Program_Chairs · 2024-01-16

Reject